# English and Co-Construction of Solidarity between Language Agents and Tourists in Tourism Information Service

## Kamaludin Yusra 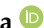

English Education Department, School of Education, University of Mataram, Mataram 83125, Indonesia; kamaludin@unram.ac.id

**Abstract:** A substantial number of studies have been completed with respect to the use of English and social solidarity in broader contexts of cross-cultural communications including tourist–host interactions in tourism settings, but little, if any, is understood about the use of English and solidarity in hectic and tightly scheduled international airport settings. This study fills the gap by explicating how English is used by Tourist Information Center (TIC) staffs and incoming tourists at Lombok International Airport (LIA), Lombok, Indonesia, to contextually symbolize solidarity among them. Data were collected in more than a year of intensive participant and non-participant ethnographic observations of real-time interactions at the TIC in the LIA. Recordings, introspective, retrospective, and prospective interviews with the staff and the tourist respondents, as well as note takings of the contexts and the situations of communicative events, were the main means of data collection, and these data were analyzed using integration of sociological analyses of solidarity and ethnographic analyses of communicative interactions. The study elucidates ideological views on the service and explicates how speech accommodation, style convergence, code switching, and kinship terms have been employed as strategies for creating symbolic solidarity.

**Keywords:** construction; solidarity; tourism; code switching; speech accommodation





## 1. Introduction

This article discusses the use of English as a lingua franca in tourism industries, particularly in Lombok, Indonesia, where the TIC (see Figure 1) has been set up with English language agents assisting incoming tourists with tourism objects to visit and with ways of getting around the island. Information about tourism objects has been widely spread in smart tourism modes such as internet and mobile applications (Koo et al. 2013; Li et al. 2016), but selling natural tourism as well as cultural events has forced the Lombok government to include the local guest-welcoming culture as part of tourism practices, as recommended in contemporary tourism management (Cohen and Cooper 1986; Dann 1996; Jafari and Way 1994; Rázusová 2009), where tourists can feel welcomed and share solidarity with the agents and the local people. This function is served by TIC staff as cultural agents at the local airport at the gate of Lombok. While the roles of tourism institutions in promoting tourism have received much attention, the role of individual agents and the languages that they use to make tourists feel at home have not received sufficient academic attention. The study is intended to fill the gap by examining, firstly, reasons why the TIC is necessary; secondly, acts of symbolic solidarity between the agents and the tourists; and finally, features of English as means of co-constructing agent–tourist solidarity in the tourism industry.

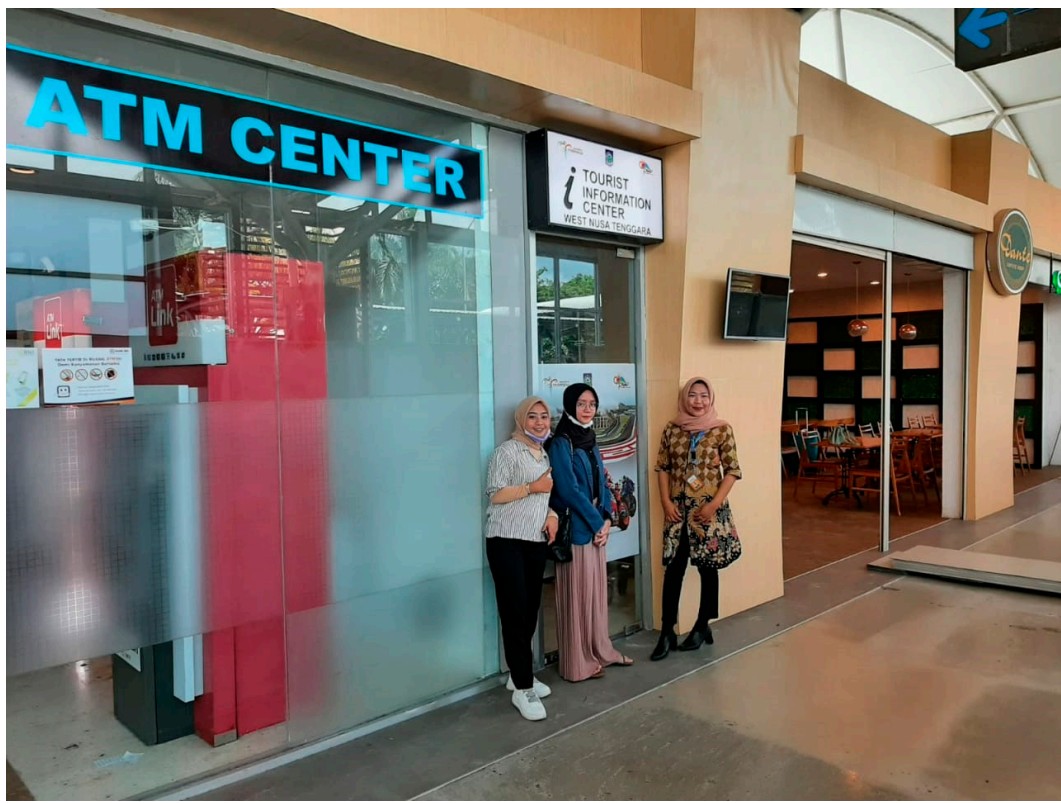

**Figure 1.** Office of Tourist Information Centre, Lombok International Airport.

Affordable airfare and safer flights have enabled massive international mobility for work, for pleasure, for family reunion, or for tourism purposes (Brons et al. 2002). The last decade has witnessed sharp increases in international mobility. Citing data from United Nations World Tourism Organization (UNWTO), Roser (2017) reported that travel mobility has not only increased in numbers (from 25 million in 1950 to 1.4 billion travelers in 2018) but also in destinations (from dominantly European in 1950 to widespread regions in 2018). Roser (2017) has also shown that from only 200 million travelers in 1970s, the number has doubled several times in 1980s, in 1990s, and in 2000. In 2018, almost 1.5 billion travelers were recorded. The destinations have similarly shifted from merely Europe in 1950 to more varied regions. In 2018, Asia-Pacific regions with 343 million visitors have replaced Americas (with 217 million visitors) as the second most visited regions after Europe (with 713 million visitors), although the number of visitors to other regions have also been increasing. In total, 67 and 64 million tourists were recorded to have visited Africa and the Middle East in that year, respectively. Statistica Research Department [SRD] (2022) has estimated that, by 2030, Asia-Pacific regions will outnumber Europe as the most favorite destination; However, as Statistica Research Department [SRD] (2022) has also shown, out of 47 billion international travelers in 2019, only 703 and 382 million of them travelled to Europe and Asia-Pacific regions, respectively.

In Indonesia, the number of visiting tourists has drastically increased from 2010 to 2019 but dropped drastically in 2020 from sixteen million visitors to only four million visitors (ASEAN Statistics 2022). ASEAN Statistics (2022) has also provided data on annual tourist visits to Indonesia in the last decade, and our preliminary analysis of these data (see Figure 2) shows that tourists of the following countries of origin have been dominant in Indonesian tourism: China (26.06%), Singapore (19.19%), Malaysia (9.54%), Korea (7.94%), Thailand (7.22%), Japan (5.86%), Australia (5.27%), US (4.53%), India (4.37%), Vietnam (4.17%), UK (3.36), Taiwan (2.86%), and Russia (2.3%). A great number of tourists with Chinese backgrounds increases the potential of Mandarin to be used as the language of interaction and, as shown in Figure 3, this potential is almost as

high as 50%. The potential of other languages is also low: the Korean language is around 20%, and the Japanese and Malay languages are around 6%. Although English is below 32%, its potential can grow to 100% due to contextual difficulties. Schedules of tourist arrivals are always unpredictable and, thus, being accommodating by providing staff who speak their languages is challenging, and consequently, English is the only available option. Even when the Malay-speaking Malaysian tourists are accounted for in the data, the potential use of English is still as high as 95%. Interestingly though, Indonesian and Malaysian speakers of the language are hardly intelligible to each other, thus the need for English increases to 100%.

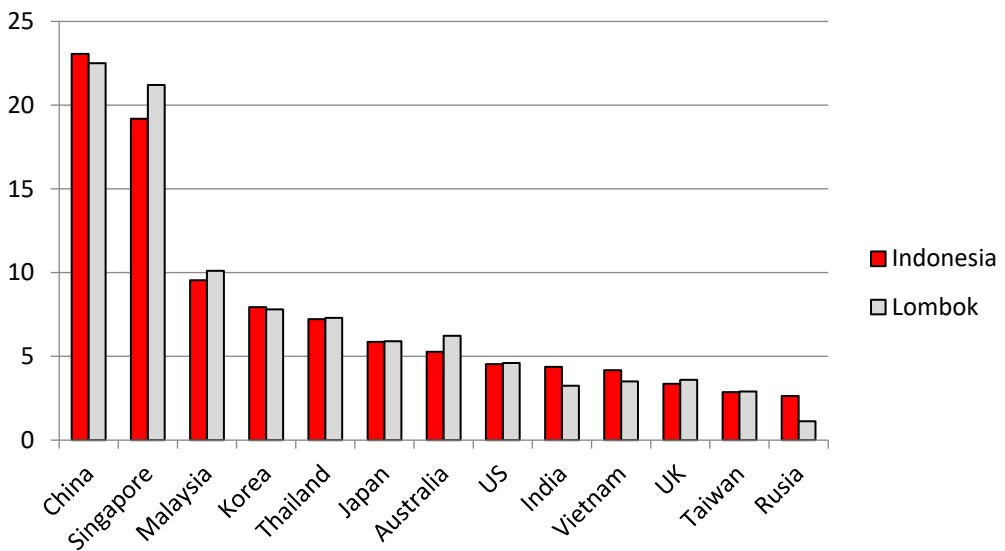

**Figure 2.** Origins and Percentages of Visitors to Indonesia and Lombok as Destinations (2011 to 2020) (Sources: ASEAN Statistics 2022 and NTB Tourism Office 2022).

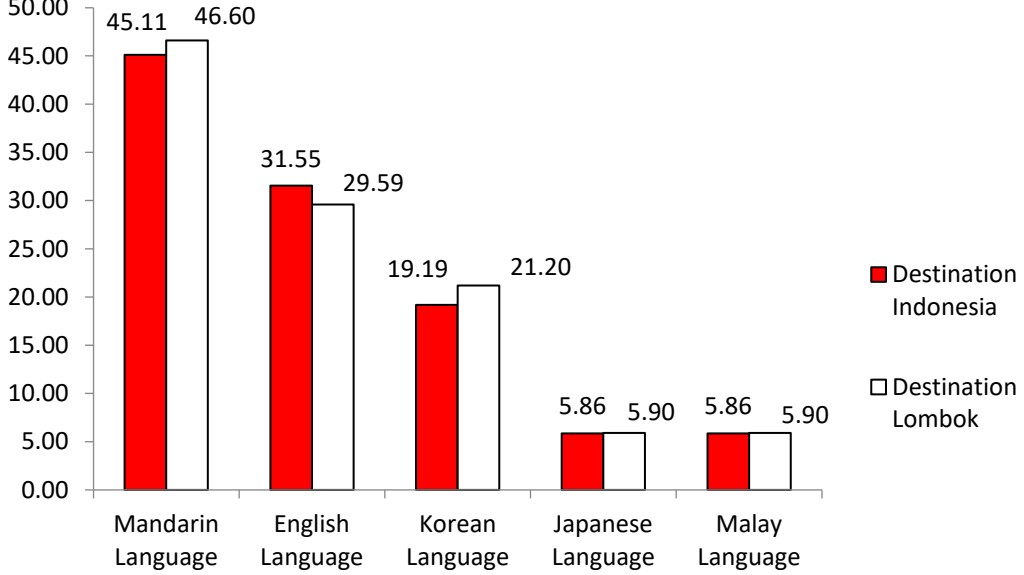

**Figure 3.** Potential Language Use Based on Tourist Backgrounds.

At the LIA, during COVID-19, domestic tourists outnumbered international ones, but after COVID-19, early in 2022, international tourists have become more dominant. In 2017 to 2021, more than 60% (N = 10,772,360) of them were domestic, but in 2022, when the COVID19 pandemic was slowly going away and tourism was increasing, around 95% (N = 541,151) of them were international (NTB Tourism Office 2022). International tourists

have once again outnumbered Indonesian-speaking ones. Lestari and Yusra (2022) have shown that, in 2022 and post COVID-19, tourists visiting Lombok are from the following countries: Singapore 22%, Malaysia 18%, Australia 11%, China 11%, Japan 9%, Korea 5%, Taiwan 4%, East Timor 3.3%, USA 3.25%, and UK 3.1%. With domestic tourists only around 4%, international tourists are more than 95%, and with average planned length of stay around 1.2 day, the local government has to establish a TIC at the local international airport that provides free emergency English language service, promoting local tourism destinations as a strategy for what Heller et al. (2014) have illustrated as creative marketing of local tourist objects and, at the same time, negotiating with them to spend more time in Lombok or to plan to revisit Lombok in their next tourism plan. In doing so, they use English language as a tourism lingua franca, following Duchĕne (2009) and Heller and Duchĕne (2016), which is also commoditized as a resource in producing and consuming local tourist objects.

Perhaps, there is a sense of emergency in the implementation. The notion of emergency here could be related to that of medical situations, although it is not as life-threating. The majority of incoming tourists are visitors (i.e., those touring less than 24 h in addition to official or business trips) and travelers (i.e., those touring longer than 24 h with personal touring interests), rather than holiday-makers (i.e., those touring for holiday, spending time and money in or around hotels); they usually set aside shorter traveling time and, thus, information about tourism objects accessible to them within their time and transport limitations will be essentially at an emergency stage (Li et al. 2020). The emergency here is not only temporal but also psychological. Jaworski and Thurlow (2010) have argued that tourism is motivated by pleasure with tourism experiences and commodities, but the dislocation of the tourists and their languages affects the visited people and languages. Being in new places and being dislocated with one's home country according to Skinner (2011) can be psychologically depressing. Thus, reliable tourism information from real people in real time will, to a certain extent, reduce the psychological tension. As Li et al. (2020) have advised us, provision of reliable information in person can bring comfort to individuals in emergency circumstances.

The context can also bring forth a linguistic emergency because with less than 1% of the local people can speak survival English (NTB Tourism Office 2022), even though the tourists urgently need information to travel around Lombok. A number of scholars have warned us that, when in new situations, people in mobility, including tourists, are concerned with the linguistic repertoires of themselves and others (Blommaert 2005), how they might operate with them given the imbalanced nature, distribution, and scale of repertoires (Blommaert 2005; Blommaert et al. 2005), and different social and moral values (Blommaert 2017) in the new multilayered situations. Thus, travelers will consider how they might construct themselves in the visited country, how the host communities might treat them, what they can or cannot do there, and whether their sociolinguistic repertoires could help them in the new situations. With limited community access to English, the provision of English service at the airport might reduce this nervousness. Additionally, scamming has been widely reported in the tourism world and such a situation puts forward a different type of emergency: tourists need to obtain information from scam-free government officials, and TIC staff might fit in correctly with this need. Finally, the emergency situation can be situated by cultural events. "Pesilaq Temui" (Sasak: guest welcoming) is one of the core values in the Sasak culture, and the service provided by TIC agents is also an emergent enactment of these values. In fact, as Jaworski and Thurlow (2010, p. 270) have strongly claimed, "the speech act of greeting has become one of the most typical resources for the enactment and mediation of tourist experience". It is on these temporal, psychological, linguistic, and cultural notions of emergency that both the staff and the tourists have developed solidarity among them. Before discussing the discursive development, perhaps, some background concepts need to be briefly clarified.

## 2. Literature Review

This article discusses how TIC staff use English to establish symbolic solidarity with the clients. While solidarity in language use among members of an established community has been widely studied, solidarity between unfamiliar individuals is limitedly discussed. This article fills this gap by looking at the use of English as a means of constructing symbolic solidarity at the airport from the perspectives of mechanical and organic solidarity in the works of Durkheim (1994) and Tönnies (2001). Following Durkheim's (1994) ideas, the article assumes that TIC staff and the tourists develop common awareness as mutual good relationship (rapport) as members of global communities, which are *Gemeinschaft* in Tönnies' (2001) conception, who similarly experience the agonies of being away from home; these feelings of being similar might invoke mechanical solidarity among them, that is, symbolic commonality among interactants due to similarities in contextual dimensions. At the same time, enacting their official roles, TIC staff can treat the tourists as official clients in need of the staff's roles and responsibilities in providing the much-needed tourism information. In this case, the TIC's main role, as one of the organs of the tourism world, is to report to the tourists, who act as other organs, the information that they need, the accomplishment of which is also reported to tourism authorities as a sign of job completion. As cultural agents, they need to position themselves as "epen gawe" (Sasak: party or homeowners) and serve the tourists as "temui pesilaqan" (Sasak: invited guests). In both ways, they complementarily enable each other as organs or networks within society, which Tönnies (2001) refers to as *Gesellschaft*, that mutually enact their roles in typical tourism language service encounters and, thus, establishing among them some sort of mutual group membership or organic solidarity (according to Durkheim's 1994 depiction of such situations). In so doing, TIC staff position themselves as marketing "producers" of tourist objects to tourists who are positioned as "consumers" (Heller and Duchěne 2012), negotiating with the customers that the native population are the "legitimate" cultural producers (Gao 2016) as they illustrate the geographic and cultural beauty of the objects (Heller et al. 2014) to the customers where everyday events are discursively made fantastic and exotic, thereby altering them from cultural purposes to revenue-generating events (Jaworski and Thurlow 2010)

Symbolic solidarity, mechanical or organic, has been investigated in a number of language service contexts, particularly in healthcare and tourism. In healthcare, solidarity has been defined as the social cohesion between displayed and host communities by creating a collective identity or sharing similar concern in response to crises (Jayakody et al. 2022). In this sense, a great body of knowledge has been recorded in the essential roles of language service agents in the provision of valid health information to patients and families with limited English competence. Kuo et al. (2007) mail surveyed 1829 American pediatrician societies and found that bilingual family members and hospital staffs had been used in healthcare to assist patients with English difficulties. Flores (2005) and Jacobs et al. (2004) found that the quality of medical interpreters play an essential role in healthcare, as it closely affects the quality of care, patient satisfaction, and healthcare outcome. Green et al. (2005) have similarly shown that the use of interpreters and doctors sharing the same background language is determinant of the provision of quality health service. The essential role that language plays in emergency response to public healthcare situations, such as the COVID19 pandemic, has made Mendoza-Dreisbach and Mendoza-Dreisbach (2021) to call for a "linguistic turn" in health service in which success or failure in providing healthcare solutions is determined by mutual sharing of languages and health information between the patients or patients' families and health service providers.

In tourism, language service is not only associated with the provision of quality information, but it is also "a source of symbolic value added to tourism attractions" and "a mode of management" (Heller and Duchěne 2012) in global networks of tourism. Cohen and Cooper (1986) have shown how language services are used as "language brokerage" in Thailand tourism, where the locals accommodated their languages to those of the tourists. Dann (1996) has highlighted the importance of language in tourism service calling it as "the

language of tourism", that is, the means through which tourism objects and experiences are visualized in such a way as to persuade readers to partake in the promised experiences. Discussing it from a sociolinguistic perspective, Dann (1996) construes that communication in the tourism world is just like everyday communication, where individuals make convergence or divergence claims, exercise power or solidarity, and contextualize personal interest or disinterest into various forms of tourism (for example, ecotourism, Muslim-friendly tourism, cultural heritage tourism, vegetarian tourism, and the like). Rázusová (2009) has highlighted that tourism practitioners are members of the same communities of practices with particular emphases on the strangeness, novelty, playfulness, and adventurousness of tourism experiences and qualifying descriptions and expressions for such dimensions should be used in language service. Interestingly, Jafari and Way (1994) and Xie et al. (2022) have called for the need to extend language service into cultural service when they suggested that tourism workers should not only talk in the languages of the guests, but they also need to treat them within their cultural expectations. Koo et al. (2013) reported that tourism language services in Korea are provided real-time using smart technologies, tourism channels, websites, SNS, and mobile applications, and in their view, smart tourism with technologically based smart-tourism language services will be the future direction of world tourism. Lestari and Yusra (2022) has illustrated that tourists visiting Lombok's cultural heritage tourism objects expected the cultural practices on offer to be performed in the native language as a way of presenting culturally authentic attractions. To a great extent, these instances exemplify the linguistic turn in tourism because language, as Heller et al. (2014) and Jaworski and Thurlow (2010) have suggested, is the only means through which a mode of industrial management in tourism can be implemented where texts and interactions are employed or commoditized for the provision, promotion, and monitoring of tourism sales.

The linguistic turn in the tourism industry has been reported from a number of geographical settings and sociolinguistic perspectives. Sharma (2018), using critical discourse perspectives while studying Nepali tourism in the Himalayas, has illustrated how porters and trekking guides have used English as a transactional means of conveying information, as an interpersonal means of building close personal relations with guided tourists, and as an economic means of commoditizing local identities and cultures into tourism markets. Using interviews and ethnography as data collection instruments, Sharma showed how the workers have made use of their English-speaking skills as an agentive linguistic means of positioning themselves in ethnic, economic and professional backgrounds relative to those of the tourists as linguistic and financial resources. Additionally, English has been used as an empowering tool for tourism workers to travel to locations that they expect to be not only under the expense of the guided tourists but also with financial benefits as well. Thus, with English as a language for communication (House 2003) in the tourism industry, tourism workers can effectively respond to the challenges that the tourism world impose upon them to their own cultural and economic benefits.

In Lombok tourism, the need for accommodating the visitors' languages, cultures, personal interest, and smart technology has been well addressed in the local practices of tourism industry. Yet, as part of the local culture, human agents ('Pesilaq' (Sasak language: guest attendant)) should be appointed to stand at the gate of the house and welcome all incoming guests to the party of the house owner. This cultural practice has been translated into the local tourism policy where TIC staff should be appointed at the TIC of airports and seaports, which act as the gates of the island. By promoting local objects to incoming tourists, TIC staff are putting these objects onto the global maps of tourism, a role usually assumed by inflight magazines (Thurlow and Jaworski 2003). Thus, the discourse between the staff and the tourists, following Heller and Duchěne (2016) and Heller et al. (2014), are no longer discourses on what are the right things to do in the contexts, but rather what cultural activities can be mobilized as new products, assets, or brands and which segments of tourists are a potential for new or niche markets. In fact, Thurlow and Jaworski (2003) have requested that tourism marketing is a matter of strategic differentiation and

promotion for commercial rebranding. This article looks into how TIC staff at the LIA enact their linguistic and cultural roles and how they perform mechanical and organic solidarity with tourists. While the role of languages in solidarity has been widely investigated in long-established multicultural settings, the role of English in symbolic solidarity in emergency-tourism airport situations has not been investigated. This article will fill the gap by studying interactions between TIC staff and tourists at the LIA. Below, the epistemology and the methodology of the study are explicated.

## 3. Materials and Methods

The study identifies the use of English by the staff and tourists at the TIC in the LIA and how this use contextually symbolizes conviviality and solidarity among them. The following pseudo-named staff from the provincial tourism office and with language skills in addition to English have been scheduled to work at the TIC assisting international tourists in English or in other languages: (1) Arif, male, English and Japanese; (2) Eva, female, English and Mandarin; (3) Danti, female, English and Korean; and (4) Noni, female, English and German. Access to the venue for data collection was guaranteed with written permits from tourism and airport authorities. Data were collected through interviews, observations, and by distributing questionnaires. Interviews and questionnaires were distributed to four tourism authorities, all TIC staff, and thirty-five tourists (7 L1, 12 L2 and 20 LX speakers of English). Note that L1, L2, and LX here refer to speakers of English as a first, second, and foreign language, respectively, judged mainly based on their countries of origin. For the staff and authorities, the questions were posed in Bahasa Indonesia, the national language, while for the tourists, the attached English versions were administered, and English was used. Introspective interviews (see questions 1 to 4 in Appendices A and B, and questions 7 to 9 and 11 to 14 in Appendix B) focused on the respondents' personal general opinions about tourism policies, service, and experiences. Retrospective questions centered on past tourism episodes, and these questions varied from context to context, depending on the nature of the tourism incidents concerned. Prospective interviews concentrated on future potential tourism undertakings (see question 5 in Appendix A, and questions 5, 10, and 15 in Appendix B). Intensive participant and non-participant ethnographic observations were conducted from January 2022 to November 2022. Participant modes were performed by helping the staff with TIC service and by becoming drivers, while non-participant roles were implemented by standing by the settings taking the roles as colleagues, cleaning service staffs, tourists queuing for the same service, passengers looking onto the language service encounters, and assistant to drivers. Audio and video recordings; introspective, retrospective, and prospective interviews with TIC staff and tourist respondents; and note takings of the contexts and the situations of the recorded communicative events were the main means of data collection. Permission to record interactions for official and research purposes were orally required from the tourist respondents prior to recording, and the majority had no objections with this. Around 150 h of interactions were recorded, and 75 tourists filled in the post-encounter questionnaire (see Appendix B).

Data were analyzed quantitatively with descriptive statistics and qualitatively with impressionistic linguistic analyses by integrating sociological analyses of solidarity and ethnographic analyses of communicative interactions in an interactional sociolinguistic (IS) analysis, which Canagarajah (2020) attributes as being most useful for analyzing transnational and translingual practices, such as tourism language services at international airports. Used as a bridge between cultural ethnography and conversation analysis, the analysis starts with identification of sociocultural contexts, within which the interactions are intended to be contextualized. With linguistic and non-linguistic cues, we identified contextual meanings, as well as other meanings, that have been recontextualized or entextualized in these contexts (Silverstein 2019, p. 56). With conversation analysis, the study inferred how language behaviors that the staff and tourists indexed symbolic solidarity among them. With Chi-square analysis, the study contrasted dimensions including accommodation of speech among the participants.

## 4. Results

This article concerns the ideological reasons behind the operation of TICs, the roles of these ideologies on the service behaviors of TIC staff, and the language features that they use for agent–tourist solidarity construction.

### 4.1. Ideological Reasons

The analysis of the interviews with tourism authorities and TIC staff as well as of the brochures distributed to interested tourists have enabled the study to tease a number of reasons for establishing the service. These reasons are ideological because ideology, as Blommaert (2005, p. 161) has defined it, is an amalgam set of rationalized ideas about better ways of conducts accumulating from history of experiences, and the service of concern here is tourism as one of government's public amenities. Extracts of these interviews have been selected below based on their relevance with ideas about how tourism should be better served. At the macro level, there is a political reason why the provision of information, comfort, and cultural practices at the airport has been one of the main roles of the global modern government. Not only does it serve the bioeconomic requirements of individuals, but the provision of information, in times of need, has also been the top priority of every modern government. The government of Lombok (GOL) and Lombok Tourism Authority (LTA) are no different; by setting tourism as their primary source of revenue and airport language service as its welcoming gate, they have to provide the best tourism service possible to invite more tourists to visit and involve Lombok and its tourist objects within what Heller et al. (2014) describe as the sociolinguistic requirement in the socio-political economy of globalized world. With a target of two million international visitors in year 2023, they have to ascertain what information about tourism objects and events is spread not only through the internet, in general, and tourism platforms, in particular, but also through direct human, face-to-face interactions. One of the managers of the LTA clearly explicates this political agenda when he says,

> Pariwisata itu sekarang menjadi agenda nasional selain bidang pertanian dan bidang lain sebagai sumber devisa. Informasi [wisata] menjadi sangat penting terutama informasi tentang festival budaya dan even-even wisata lainnya. Memang, informasi ini sudah ada di internet, surat kabar, televisi atau pun media sosial, tetapi kami berpendapat bahwa hal itu belumlah dirasa cukup. Kita perlu sentuhan manusia dimana informasi disampaikan oleh manusia dan wisatawan dapat bertanya sesuai kebutuhan mereka. Ini merupakan cara cerdas kita setelah kita sukses gemilang dengan wisata budaya, wisata reliji, wisata olahraga dengan world superbike dan MotoGP (Buldi, 50, pejabat kantor pariwisata)

> Tourism is one of our top political agenda in addition to agriculture and others for revenue generating. [Tourism] information is essential, especially, information about cultural festivals and other tourism-salable events. The information is already there on the internet, newspapers, TVs, and social media, but we think those are not enough. We need a human touch in it, offering information from human sources allowing visitors to ask questions of concern. It is, indeed, our latest 'cara cerdas' [smart innovation] after the great success of cultural tourism, religious tourism and, now, sports tourism with world superbike and motorcycle grand prix (Buldi, 50 y.o., tourism authority, my translation).

Thus, from a political point of view, TIC services are a manifestation of good government and modern tourism management—a practice shared by the majority of world's governments and tourism authorities. Like modern governments elsewhere, GOL is compelled to provide tourism structures and infrastructures, but the LTA is obliged to offer tourism information that visitors can make use of within their travelling time. Although tourism information has been transmitted through smart technologies with big mobile data (i.e., internet, social media, and mobile applications, among others) regarding the transportation, travel intermediary, hospitality, and entertainment sectors of tourism, the

LTA is assigned to create tourists-friendly modern tourism services. These ideologies are explicated, at least, in Eva's descriptions of her role in the tourism language service.

> Saya agak bisa sedikit bahasa Mandarin. Tapi, bahasa Inggris yang saya bisa. Itu mungkin sebabnya saya ditugaskan melayani turis dari Cina, Singapura atau Taiwan. Bagus sih ndak, saya dalam bahasa Mandarin, tapi untungnya mereka datangnya berkelompok dan ada ketuanya dan bisa bahasa Inggris. Sudah sih dia rencanakan apa-apa dia mau lihat, tugas kita jaq beri tambahan informasi, terutama even-even baru yang belum ada di internet. Umumnya dia Tanya tentang pantaipantai yang sudah terkenal, Senggigi, Kuta dan Gili Trawangan, tapi kita imbuhi dengan even-even baru disana yang bisa mungkin mereka nikmati (Eva, 48, petugas TIC)

> I can speak a bit of Mandarin. However, English is my strength. So, my job is to serve Chinese [speaking] visitors from China, Singapore, or Taiwan. I am not that good in Mandarin, but luckily, they come in groups and a leader with English skill. Of course, they have planned what to see from the internet, [but] my job is to give them more choices, especially those that are not on the net yet. Usually they ask about famous beaches like Senggigi, Kuta and Gili Trawangan, but we offer what events are there for them to enjoy (Eva, 48, TIC staff, my translation).

Thus, the official role is enacted here not only by answering tourism questions but also by promoting newly established events that might be of interest to visitors.

In most cases, the official role was enacted in culturally traditional ways; by dressing in traditional costumes and accompanied occasionally with Sasak *gamelan* music, the staff enacted their role as guest escorts called "penyilaq", they greeted tourists with the Sasak welcoming verbal expressions "sugeng rawuh" (Sasak: welcome), "salam rawuh" (Sasak: welcome" and with the English expression "Welcome to Lombok" accompanied with a right-hand gesture pointing to the direction of TIC, the welcoming staff enacted their cultural behaviors and values in guest welcoming. In general, the right-hand thumb points to the direction of the TIC as the staff expects visitors to make use of TIC services, and thus, the staff could explicate more about local tourism. This cultural awareness in the job performance was mutually shared among the staff and music players. When asked about her cultural role, Noni, the youngest member of the team, mentioned that the visitors were usually suspicious of them and thought they were just like others in the area disturbingly offering service to unwanted tourists, but with the strikingly different nature of the staff's uniforms, costumes, personal identification card, and with a bit of personal introduction, persuasion, and culturally polite manners, they were appreciative of the service. Describing the situation in the service, she said,

> Umumnya, mereka terkesima dengan bantuan seperti ini. Banyak yang nanya-nanya tentang atraksi-atraksi budaya. Laguq, mereka juga nanya hal-hal sepele juga. Misalnya, 'Where do we get a bus to town?' Dan, kita tinggal tunjukin saja tempat jemput penumpang dan bilang''It is out there as soon as you get to the welcoming hall, it is the big blue bus on your left'. Klo santai, mereka mau dengar kita dan kita bisa kasih brosur atau informasi festival dan atraksi budaya yang ada. Waktu acara WSBK dan MotoGP, agak lebih sulit. Kita layani juga penonton lokal (Noni, 21, petugas TIC).

> They were usually surprised to have assistance like that. Many asked for questions specific about tourism attractions [that they can visit]. However, they also asked for simple things like 'Where do we get a bus to town?' Additionally, we just need to point to the passenger picking up hall and say 'It is out there as soon as you get to the welcoming hall, it is the big blue bus on your left'. When relaxed, they would listen to us and we could give them the brochures or inform them of upcoming cultural festivals and attractions. During the WSBK and MotoGP events, things got tougher [because] we also served the domestic spectators (Noni, 21, TIC staff, my translation).

Out of the 227 visitors observed during the fieldwork, we informally interviewed 35 of them about what they thought about the service, and the majority of them saw it as unique, surprising, and helpful. Amir and Dagney, a holiday-making couple from the USA, reported that the service is unique, as they have never encountered such a service in all of their holiday-making travels. To Muslim tourists such as them, information that all meat used in food in Lombok is "halal" (Arabic: sacred) (Amir, 30, tourist), and all mosques are open to all Muslim wishing to pray, even those who perform non-Sunni prayers such as them, is "as precious as gold"(Dagney, 28, tourist), as such information cannot be found on the internet. Stacy and Belen, models and artists from Argentina, saw it as very astounding as they have never expected to be wonderfully welcome in such a way, even in their "glamorous life" (Stacy, 25, tourist) back home. As first-time travelers to Lombok, after several tourism experiences in Bali, the information about places to visit in Lombok that were like a "new Bali" was seen as helpful to them to make adjustment to make better tourism plans. They were very grateful for the service, as it helped them to "put new things in the menu especially the beach with pink sand [Pink Beach]" (Belen, 23, tourist), which they did not realize was near the airport. In the surveys with the questionnaire in Appendix B (items number 12 to 15), 227 tourist respondents indicated that TIC services were helpful (81%), the information provided was useful (83%) and relevant with travel needs (96%), and the TIC performance was excellent (97%). Thus, through simple and far from tech-savvy, direct face-to-face language services, TIC staff have proven themselves to be useful to the visitors and the tourism objects, presenting not only tourism information, or the "soul" (Heller et al. 2014), and technical tourism-related English skills (Gao 2016) but also commoditizing the objects with added economic values and ways of consuming them (Heller et al. 2014).

We have talked about the ideological, political, and cultural beliefs governing the behaviors in the emergency language service at the LIA's TIC. Let us now scrutinize how the ideologies symbolically shape solidarity between the staff and visitors in the language service discourse.

*4.2. Symbolic Solidarity*

In order to examine interethnic solidarity in TIC's information services, we need to look at the power and privilege dynamics between TIC staff as service providers and the tourists as service recipients. These dynamics, according to Baker-Boosamra et al. (2006), are related with honesty, reciprocity, and mutuality established and promoted between the groups. In the tourism world, Zhang and Tang (2021) describe solidarity as the feeling of being welcomed, sympathetically understood, and emotionally close between tourists and service agents.

The acts of solidarity in tourism language services can be seen in the dynamics of interactions between the staff and tourists. The staff, acting as information provider, are seen by the tourists, acting as service recipients in every context of interaction, as honest officials who can provide reliable information. Honesty, according to Hwang et al. (2022), is part of affectional solidarity. The tourists can require similar information from guides, drivers, or other information providers out of the airport, but the latter might require financial compensation for the service. The information that they provide might not be accurate or comprehensive, as it might be more beneficial to them than to the tourists. The interaction can also be for reciprocity. Reciprocity in paid or unpaid service can be categorized as consensual solidarity (Hwang et al. 2022). TIC staff need to perform their job of providing information, while the tourists need the information. In such relationships, the staff might exercise power over the tourists, as King (2004) has argued, but the reciprocal need for job performance on the part of the staff and information requirement on the part of the tourists has brought the interaction more toward a solidary relationship where each party is reciprocally involved in collaborative actions: the tourists' need for information is compensated by the staff's abundance of it. Thus, in the service interaction, both parties

mutually assist each other in acting out their roles as active social agents, and from the perspective of Hwang et al. (2022), this linkage is associational solidarity.

Honesty, reciprocity, and mutuality in service encounters facilitate the exercise of Brown and Levinson's (1987) positive politeness. This enables TIC staff and tourists to make use of positive politeness strategies, for example, by using nicknames, using of bald-on-record expressions, humor, jokes, teasing, or ridiculing (Sifianu 1992). Such playful language use can lead to symbolic solidarity between the interactants, and this can be described as mechanical and organic solidarity. Mechanical solidarity is shaped and reshaped by bodily co-presence (Johnson 2022) and contextual similarities between the TIC as the service provider and the tourists as the service recipients, in which both of them are governed by their concerns or "mutual; focus of attention" (Johnson 2022) with tourism and its related aspects (e.g., flights, accommodation, and food and beverages). In many of these interactions, conversations started normally with greetings from the visitors with mechanical solidarity-making questions such as "Are you on duty?"(Tapescript (TS) 3 line 8) or "Is this the tourism information center?" (TS 5 line 11) or "Is the information service free?" (TS 4 line 2) which were responded to with affirmative answers like "Yes and what can I do for you?"(TS 4 line 3) or "yes, what can we do for you sir? (TS 5 line 12) from the staff. Some forms of mechanical solidarity were also inserted in exchanges where friendly expressions, such as "How was your flight?" (TS 11 line 6) or "First time to Lombok?" (TS 6 line 14), and such expressions helped them in sharing the symbolic associational mood (Johnson 2022). Some forms of close relationships were also established in personal introductions, usually at the beginning of the exchange, when tourists mention their country of origin, and the staffs exclaim, "Oh I have been there" (TS 10 line 8), "Oh I love that place", or "Oh I would love to be there one day" (TS 2 line 125), which was also affirmatively responded to by the tourists in rather mechanical manners such as "Oh, did you like it?" (TS 10 line 28) or "Oh come, it is a nice place" (TS 3 line 75). To a great extent, these data indicate a certain degree of formality, but the shifting away from the formal discourse of tourist destination to personal matters, experiences and interests is a significant reduction from a formal to a personal, informal, and solidary relationship. Extract 1 exemplifies this solidarity.

**Extract 1: The Ironman (Nancy and Steve LN168-182)**

*Eva, a TIC staff member, is handing over Australian tourists Nancy and Steve to Yadi, a private-car driver, to take them to the Sheraton Hotel in Senggigi, Lombok.*

(1)  Yadi: come with me
(2)  Eva: it is Yadi, your driver
(3)  Man: Steve
(4)  Lady: Nancy
(5)  Yadi: the car is over here
(6)  Nancy: Thank you very much for your help
(7)  Eva: it is my pleasure
(8)  Steve: Thank you. Sorry, I did not ask you earlier. What's your name?
(9)  Eva: Eva
(10)  Steve: Thank you for your help
(11)  Eva: pleasure helping you sir
(12)  Nancy: Much appreciated Eva
(13)  Eva: Enjoy your holiday in Lombok
(14)  Nancy: thank you, bye
(15)  Eva: bye

Nancy and Steve, arriving from Bali, were in an emergency situation; they were trying to find ways to get into the Sheraton Hotel in Senggigi, but the main road from the airport was closed due to the Ironman championship. Eva, the staff member, helped them. She found that all Sheraton pick-up cars were stranded between the road blockage, and she had to find alternative ways to assist the tourists. Yadi, a private-car driver, was called to

take them to the hotel via alternative roads. In lines (1)–(4), they introduced themselves. In line (5), Yadi the driver took the tourists to the car, and Nancy, in lines (6) and (12), used the opportunity to appreciate Eva's help, and in lines (8) and (10), Steve did the same. Eva responded to them in familiar expressions in lines (7), (9), and (11). In line (13), Eva bid farewell to Nancy in a rather formal manner wishing her an enjoyable tour of Lombok. In lines (14) and (15), both the tourist and the agent departed in informal greeting saying only "thank you" and "bye". Such expressions are usually shared only among friends sharing similarly common interest (Jucker 2017) and are formulaic in tourist–host interactions (Jaworski and Thurlow 2010); for these reasons, it exemplifies mechanical solidarity. In formal host–tourist interactions, a more formal welcome would have been opted for, for example, by saying off-record negative politeness expression "Welcome to Lombok, Madam" or "Enjoy your time in Lombok, Madam" with falling intonation, rather than in a bald-on-record positive politeness expression such as "Enjoy your holiday in Lombok" and in raising tone.

Organic solidarity, on the contrary, is established in the functioning of TIC staff as the service providers and the tourists as service recipients. The presence of international tourists, on the one hand, is a sign of solidary international support for the local socio-economic context, but the presence of the staff as information providers is also a token of solidarity with the visitors—that they will not feel alone in their adventure into an unknown world as friends like the staff are always available to provide a helping hand. In the words of Danti, one of the staff, the tourists will not be "leger laloq" (English: very nervous) being in a strange land such as Lombok, and in the words of Naresh, an Indian solo traveler, the service gives him "a peace of mind" that he is at least in the right place for his new (surfing) adventure. Such solidary support was also felt strongly by the locals and by the tourists who decided to travel while others decided to stay at home due to COVID19.

Through solidarity, nonetheless, the tourists as the recipients will always determine the types of information that they require. Once informed, they can serve themselves with actions. TIC staff as the service provider can learn from language service experiences, and they have control over what and how to learn from the experiences (Baker-Boosamra et al. 2006). In the 227 observations conducted during the field work, 204 services were initiated by the tourists, and only 23 were initiated by the staff. The types of service that they required were related to access to hotels (around 30%), upcoming cultural festivals (around 25%), about particular tourism objects (around 20%), transport to the city (around 15%), and access to tourism objects (around 5%). There were instances where service ideologies between the parties were contested. The staff was intrigued by their official tasks and they, thus, initiated the service as enactment of their official role. At the same time, the tourists in urgent need of information usually took the initiative and controlled interactions to suit their needs. Learning from experience, the staff followed the recipients' requirements by providing information that the recipients required. In a way, this is a form of discursive solidarity, as it is indicated in the extract below.

**Extract 2: The WSBK (Safira and Najib LN8-LN22)**

*Noni, a TIC staff member, is serving Malaysian husband–wife tourists Safira and Najib, instructing them on how to get to their hotel in Senggigi while escaping road closures during the Ironmen championship.*

(1)  Noni: Do you want to go directly to Senggigi?
(2)  Safira: yes to Senggigi
(3)  Noni: or you want to go to go to Sekotong?
(4)  Safira: We just want to get into Senggigi
(5)  Noni: Oh yes Senggigi ok
(6)  Najib: We need to check in a hotel in Senggigi
(7)  Noni: Which hotel sir?
(8)  Safira: Well we just want to know how we can get into Senggigi

(9)  Noni: I see madam
(10) Najib: Yeah Senggigi. The Kila Sengigi Hotel.

Extract 2 exemplifies such conflict of interest. In lines (1) and (3), Noni, the staff member, expected to perform her job by informing the tourists about tourism objects in Sekotong, which was referred to in exchanges prior to the extract. In a retrospective interview, Noni confessed that her question in line (1) was motivated by her intention to introduce tourist objects in a place called Sekotong as an additional destination to the list of objects that the couple was planning to visit. This exemplifies what Heller and Duchěne (2012) describe as a producer–consumer interaction, where language is used as a means of presenting tourist objects not as pieces of information but, rather as tourism products being put on sale. At the same time, the tourists in urgent need of information usually took the initiative and controlled interactions to suit their needs. In lines (4), (6), (8) and (10), both Safira and Najib the tourists were only interested in the information that they required at that moment, that in, how to go to Senggigi from the airport crowded with passengers coming in for the World Super Bike event. Learning from experiences that tourists are only concerned with what they need, Noni fulfilled their expectation by informing them how to get to the pick-up bus from the hotel. Though conflicting at its face value, the interaction indicates symbolic solidarity where both the staff and the tourists symbolically co-constructed their discursive roles.

Symbolic solidarity is existential not only at macro discursive levels but also at micro linguistic levels and this is exemplified below.

### 4.3. English and Symbolic Solidarity

Solidarity between the staff and tourists can also be symbolized in the use of verbal and non-verbal expressions in English. To a great extent, these expressions can be treated as features of symbolic solidarity. Non-linguistic features in the forms of facial and bodily expressions can also be found, but the most frequently used features are the linguistic ones in the form of style accommodation, code switching, and kin terms.

The most apparent symbolic exercise of solidarity between tourists and TIC staff was in the form of speech accommodation, in which L1 and L2 speakers of English slowed down their speech when the listeners are Lx speakers of English. This solidarity-making strategy can be perceived in Table 1.

**Table 1.** Speech Accommodation based on wpm.

| SPEAKER | LISTENER | | |
|---|---|---|---|
| | **L1** | **L2** | **LX** |
| L1 | 170 | 142 ** | 121 * |
| L2 | 144 ** | 136 | 118 ** |
| LX | 122 * | 120 ** | 111 |

Note: * Significant difference, ** no significant difference.

Table 1 shows L1 and L2 speakers accommodating their speech to that of Lx speakers of English such as TIC staff (and the drivers). L1-English-speaking tourists slowed down their speaking speed from normal to slow speech. In L1–L1 conversations, our data show that L1 speakers on average held 69 min conversations and produced 11727 words, and this indicates that the average is around 170 wpm, counted based on the number of words divided by the minute length of speaking by each L1 speaker. This can be treated as the normal speech rate in L1–L1 interactions. With L2 speakers, the speech rates range from 139 to 155 words per minute with an average of 143 wpm, calculated based on the total number words produced (i.e., 151, 151 words) divided by the minute length of conversation (i.e., 1057 min). Thus, L1 speakers of English reduced their speech by 26 wpm when interacting with L2 speakers, This speech accommodation is found to be consistent with those of

other studies in the field ([Baese-Berk and Morrill 2015](); [Morrill et al. 2016]()). However, with Chi-square analysis, the study finds that the difference in speech rate is not significant in a one-tailed test ($p > 0.05$) indicating the absence of L1–L2 speech accommodation. However, with Lx speakers, the speech rate is between 120 to 124 words per minute (average: 122 wpm). With Chi-square analysis, this speech accommodation is found to be highly significant ($p < 0.001$). By the same token, L2 speakers, when speaking among themselves, have speech rates between 129 and 147 words per minute (average: 136 wpm), but with Lx speakers of English, the speech rates are between 101 and 126 words per minute (average: 119 wpm). With Chi-square, the analysis found no significant difference ($p > 0.05$). An analysis was also performed on Lx speakers to L1 and L2 listeners, but no significant difference was found. Thus, speech accommodation was found to be significantly made only by L1 speakers.

One might argue that the accommodation is due to individual idiolects. Note, however, that the individual speech variations have been moderated with the significant number of speaker samples (N = 35) and length of interactions (more than 80 h and 23,808 words) used as the basis of the quantitative analysis above. Most importantly, the comparative analyses with Chi-square above strongly indicate the accommodation of English speech rate between L1/L2 and Lx speakers but not between L1 and L2 speakers. These statistical differences cannot be attributed to individual ways of speaking because the analyses have controlled for individual speaking variations by using the mean numbers of words per minute per individuals in various contexts of speaking in the analyses: L1–L1, L1–L2, L1–Lx, L2–L2, L2–Lx, and Lx–Lx. With such complex contextual varieties of data sources, the speech accommodation discussed above might not be attributable to the individual ways of speaking of the samples.

These this considered, these inferential statistical findings are supported by the participants as empirical sources of data. The slowing down in the speech of the L1 English speakers, according to L1 speakers, is an index of symbolic solidarity with TIC staff as Lx speakers of English might find it difficult to get the message when it is conveyed in a normal conversational speed. In the words of L1-English-speaking tourists, slow speech is used to let staff "hear correctly" (Jo, 37, US) what information is needed or to get each other to "click in the interaction" (Ela, 48, UK), as the staff and tourists may not been in tune with each other's speech tones. Nonetheless, the slow rate in the speech of TIC staff does not only indicate their status as LX English speakers, but it is also a symbol of linguistic solidarity with the tourist clients who need to comprehend the information clearly, and thus, conveying it in comprehensible rates is a sign of solidarity with the tourists and their needs. To a new staff member such as Noni, slow speech is required to compensate for her worries about making grammatical mistakes, but to experienced staff such as Arif and Danti, a slow rate of speech was used to highlight their competence in the English-speaking style of the interacting tourists, while for fast-speaking and senior staff such as Eva, the speech rate highlights her success in learning English by speaking at a "normal" speed, or "kayak bule" (like L1 speaker), and as naturally as an L1 speaker of English. These indicate that the acts of speaking in these settings symbolically imply not only the need for information from organic units of the tourism industry (i.e., organic solidarity) but also the status of each speaker in the communities of English speakers (i.e., mechanical solidarity). Thus, the interaction in itself is an act of symbolic solidarity.

Symbolic solidarity can also be seen through the accommodation of speech styles. The great majority of tourists with and L1 and L2 English background adopted the international variety of English, neglecting the Inner Circle variety of English to a certain extent. Tourists of American background were oftentimes found to reduce the postvocalic /r/ sounds ([Labov 1986]()), although the linguistic identity can still be found in my data. The same also happens to those of British background who reduced the glotalization of /t/ in middle position to a minimum, although the linguistic identity could be still found. According to [Clyne et al.]() ([2001]()), the Australians and the New Zealanders have been known for these ethnolects, but they reduced the occurrence of these broad diphthongs /aI/ and /oU/,

and the open low vowel /a/ when involved in out-group interactions. While knowing the ethnic backgrounds of their counterparts, the TIC staff as LX speakers of English, on the contrary, tried their best at showing off their English to the tourists by picking up their speech styles and leaving the international variety of English, to which they have been exposed in their education and training backgrounds. Although slightly unwanted to L1-English-speaking tourists (indicated by their pausing, changing the way of speaking, or stopping the conversation), the staff, in a retrospective interview, reported of using the style as a "penghargaan" (Indonesian: a sign of respect), a "tanda akrab" (Indonesian: sign of closeness), or a "maksud baik" (Indonesian: sign of positive gesture) to show that they were very exultant with their visit and wanted them to feel comfortable in Lombok—as if they were at a "balen mesak" (Sasak: owned home). Let us look at this practice in Extract 3, a conversation between Aflex and Anne, Australian tourists, and Danti, a TIC staff member.

**Extract 3: Route to Gili Trawangan (Aflex, Anne and Danti LN19-LN28)**

*Aflex and Anne, Autralian tourists, are inquiring about the fastest way to get to Gili Trawangan. Danti, a TIC staff member, is assisting them.*

(1) Danti: Are you going directly to Gili Trawangan?
(2) Aflex: **No** [na$^w$u/. We want to check in the hotel.
(3) Danti: Which hotel?
(4) Anne: We will be **staying** /sta$^y$ing/ at Medana Hotel.
(5) Aflex: We have been **told** /ta$^w$uld/ it is **close** /kla$^w$us/ to Gili Trawangan
(6) Danti: Yes, it is **close**/kla$^w$us/. It is only around 15 minutes' drive from your hotel.
(7) Anne: ok. Pretty **close** /kla$^w$us/.
(8) Danti: yes madam, you can call a private car from here.
(9) Aflex: so the road is not **closed** /klo$^w$ust/?
(10) Danti: it is **closed** /kla$^w$ust/. The driver **knows** /na$^w$uz/ the alternative **way** /waI/.

Though conflicting with the tourists' interest, the style accommodation by the staff is also evidence of linguistic solidarity. In line (2), Aflex, an Australian traveler, responded to Danti's question in line (1). Using a lowered /au/ diphthong reflected his identity as Australian, and this linguistic identity is strengthened with his wife's linguistic identity in line (4), who produced a diphthong with an open and lower tongue position, producing /sta$^y$ing/ rather than /ste$^y$ing/ as in mainstream Inner Circle speakers of English. This linguistic identity is again rearticulated in line (5), where Aflex once again reproduced the linguistic identity with /ta$^w$uld/ and /kla$^w$us/ and not /told/ and **close** /klo$^w$us/ as usually articulated in other Inner Circle speeches. In line (6), the staff member took up the clients' accent and articulated "close" as /kla$^w$us/ similar to Anne's (line 7). Frowning and looking at Danti, Aflex was worried about his speech style as it might cause communication failure and in line (9) he returned to the international variety of English and reshaped "closed" as /klo$^w$ust/. Danti, the staff member, failed to recognize Aflex's concern with his style, and in line (10), she continued with the Australian accent. On the way to the pick-up station, Aflex questioned Danti's Australian accent, and Danti's response was that her ability might have been accumulated from her frequent contact in the job with Australian tourists. Thus, her accent was a sign of her closeness to Australian tourists whom she saw as the most "ramah" (Indonesian: friendly) tourists of all. In this case, the speech accommodation described above is, again, a sign of symbolic solidarity.

Another interesting sign of symbolic solidarity is through the use of code switching, particularly in discourses involving Malay-speaking tourists. Tourists of Singapore, Malaysia, Brunei, Pataya (Thailand), and East Timor backgrounds have the capacity to speak either in English or in the Malay language. Although English is preferred for political and linguistic reasons, the Malay language was also inserted, particularly among Malaysian and Bruneian tourists, but they switched to English when discrepancies in meaning existed in the words selected. This is exemplified in Extract 4.

**Extract 4: Rembiga Satay (Nur and Arif, LN6-LN15)**

*Nur, a young Malaysian traveler, was looking for a hotel to stay at. It should be near Senggigi Beach where she could find a new food type called Sate Rembiga.*

(1)    Nur: What do you call this (pointing to a picture)?
(2)    Arif: sate rembiga/*Rembiga Satay*
(3)    Nur: satai rembiga ni/*This Rembiga Satay*
(4)    Arif: iya, it is satay from beef
(5)    Nur: sadapkah? Delicious-kah?/*Is it delicious*?
(6)    Arif: enak/*delicious*. Delicious. Very spicy.
(7)    Nur: yes. It looks delicious. Awak mahu/*I want it*. Tapi, di Senggigi adakah?/*But, can we find it in Sengggi?*
(8)    Arif: No, it is only in Rembiga.
(9)    Nur: Ah, so, Rembiga is the name of a place?

Malay–English switching is associated with potential misunderstanding due difference in word meanings. As shown in Extract 4, Nur, in line (5), used a Malaysian expression which means differently when seen from the Indonesia meaning. To Malaysian speakers, "delicious" is expressed in the word "sadap" (Malaysian: delicious) but in the Indonesian variety this word means "tap" or "secretly tap someone's telephone". Even when it is uttered in the Indonesian variety "sedap" (Indonesian: delicious), it is only used as expression of exclamation after having a delicious meal and not for asking the taste of prospective foods; in the latter context, the word is "enak" (Indonesian: delicious). In many cases, the use of the Malay language led to confusion for both parties, and switching to English was an effective strategy; however, the use of Malay, accentuating identities associated with the language (Mapelli 2019; Moustaoui Srhir et al. 2019), might minimize social differentiation and increase solidarity (Bhatt and Bolonyai 2022) between the staff and clients as members of the same speech community, although they opt for English to avoid possible misunderstandings.

Finally, the use of fake kinship terms conveys some sort of familial solidarity between the staff and clients. To senior holiday makers, the language crew have been trained to address clients politely with respect address forms such as *sir*, *madam*, or even *mister* or *miss*. To younger travelers, expressions such as *brother*, *sister*, *bro*, and *sis* were used. Note that addressing guests with respectful or solidary forms of address is also part of the enactment of cultural values and roles in the traditions of "pesilaq temui" (Sasak: guest invitation). Having no kinship relations, the staff's use of these terms signifies the construction of a familial relationship, and this, in turn, is also a sign of symbolic solidarity.

**Extract 5: Motorcycle (Silvio, Bobby, Danti & Arif LN35-LN44)**

*Silvio and his traveling friend, Bobby, from East Timor, are consulting Dianti and Arif about the cheapest transport to travel around Lombok.*

(1)    Silvio: Just to see around the island bro
(2)    Arif: You'd better rent a car bro
(3)    Silvio: A car? Is not expensive bro?
(4)    Danti: For one day, it is around Rp 450 thousands
(5)    Arif: yes, it is better than taxi or public transport
(6)    Silvio: Really? What do you think bro (to Bobby)
(7)    Bobby: e um preco justo (it is a fair price)

It is typical of tourists of particular ethnic and age backgrounds (Malaysian and East Timorese) to use kinship terms to address staff, and the staff oftentimes follow suit. The use of kinship terms in Extract 5 was initiated by Silvio's address to Danti with "sister" before he used "brother" with Arif in line (1), which the latter responded likewise in line (2). In line (6), he used the form to call his travel partner. The kinship terms "brother", "bro", "sister", and "sis" were also used when they greeted farewell to each other. Perhaps, similarity in age as young persons might have facilitated this contextual solidarity.

Having discussed ideological reasons, symbolic solidarity, and solidarity-making strategies, let us now compare these findings with others from studies in other contexts.

## 5. Discussion

This study has shown the ideological reasons, the service behaviors, and the intercultural solidarity enacted through TIC services.

The tourism language service has been politically and culturally motivated by the need to provide better service in tourism. Such emergency practices are not actually new, and other countries of the world have done likewise. Zheng et al. (2015) has shown a similar effort of serving tourists by the Beijing government, where staff, street signs, and tourism brochures have been written in simplified Chinese characters, in English, and Latin letters in order to accommodate the needs of oversea Chinese and general tourists. This political and cultural motivation has also been shared in the current study, but the difference is that, in Lombok, human face-to-face services have been provided in addition to smart mobile internet service, as real human interaction is preferred, as it has been an essential part of the local cultural friendliness. In fact, as shown above, the service has been framed within the local cultural norms, where tourists, as "temui pesilaqan" (Sasak: invited guests) must be culturally treated in the best quality service possible, and this has been the core of the Sasak cultural values called "tindih" (Sasak: good manner) (Nuriadi 2021). Not only do TIC staff help tourists with language difficulties when communicating with local people with no English competence, they also provide tourism information upon arrival. Such practices actually treat tourists both in their own (Jafari and Way 1994) and in local cultural frames (Lestari and Yusra 2022) in real-time interactions (Koo et al. 2013), and negotiate, during these interactions, (Heller and Duchĕne 2012) local and global neoliberal identities (Gao 2017, 2018), and social boundaries and differences (Jaworski and Thurlow 2010). In this way, they might have also enhanced the quality of tourism handling, increased tourist satisfaction, increased the length of stays, and raised the possibility of revisits, and these are worthy of further investigation. By using English not only as a lingua franca (House 2003) but also as a means for cultural and economic benefits (Sharma 2018), the staff have actually exemplified the linguistic turn in tourism world in Indonesian settings.

The study has also exemplified mechanical and organic solidarity between the staff and tourists. In a number of cases, both the staff and tourists have revealed some sort of solidarity among themselves due to reciprocal, mutual needs. Tourists, who are in a foreign land such as Lombok, require trustworthy friends that they can rely on and, as Pearce (2011) has argued, the only reliable free-from-scam information providers are government officials, and all TIC staff are government officials working inside official premises. Tourists who are new to Lombok need information about tourism objects to visit and the staff possessing the information are officially enforced to share the information freely and honestly. Both the tourists and staff mutually establish similarities among themselves as co-members of tourism communities in those particular discursive dimensions. This is essentially what Durkheim (1994) described as mechanical solidarity, which Tönnies (2001) ascribed as communities of contextual similarities; however, the study has also instantiated the workings of TIC staff and tourists as networks of organs in the tourism world, where the former serve as providers and the latter as recipients of the information service. By indirectly paying for the service, the latter initiate interactions requiring information, and by being indirectly paid as government officials, the former's responses, as Baker-Boosamra et al. 2006) has also demonstrated, are controlled by the latter. Co-constructing these roles, the staff and tourists have formed organic solidarity (Durkheim 1994) or communities by reciprocal association (Tönnies 2001) among them.

This study has also exposed how English used in these interactions symbolize solidarity. A reduced speed of speaking by the tourists is, in essence, solidary accommodation to the English ability of the staff as LX speakers of English. A slow rate of speech is also employed by the staff as a means of clearly conveying tourism messages, but at the same time, it allows them to compensate for weaknesses in English or even highlight their En-

glish competence by taking up correct expressions or preferred English accents. As Jafari and Way (1994) have advocated, using the client's culture and accent is preferable in the tourism world, as it indicates closeness and honesty among them. The dimensions of closeness and honesty are also stressed in the use of a language (i.e., the Malay language) that is shared by the staff and tourists, and this is another form of mechanical solidarity. However, for political or cultural reasons, switching the language to English is also another form of organic solidarity in the tourism word, where English is the solidarity language where political issues and possible misunderstandings are set aside and organic solidarity is constructed. As Sczepurek et al. (2022) has implied, code switching is socio-politically motivated, and social cohesion is also one of the motives. The use of solidarity kin terms also symbolically strengthens the solidarity feelings to a family-like relationship.

Now that the findings have been discussed, it is necessary to summarize the key points in a brief conclusion.

## 6. Conclusions

The study has illustrated how English is used as a means of constructing symbolic solidarity between TIC staff as service providers and international tourists as service recipients. The study has also revealed that solidarity is not only contextually enacted but also motivated ideologically by the need to perform technologically smart and culturally humane tourism management, wherein modern technologies and culturally local human touches are mutually exploited. With this approach, the local government has accentuated mechanical and organic solidarity among the tourists, staff, and the people of Lombok because tourism and the tourism industry have been brought to the political, social, cultural, and individual levels. Various linguistic forms have been utilized as solidarity-making strategies: the exploitation of slower rates of speech, particularly among L1 tourist speakers of English; the staffs' converging to English-speaking styles of the tourist clients; Malay–English code switching between the staff and Malay-speaking tourists; and the use of pseudo-kinship terms, particularly by tourists of equal age to the staff. The use of these strategies has indeed brought interactions in the tourism world to more solidary intimate human-to-human levels. Nonetheless, the impact that this tourist-friendly service has on tourists' satisfaction and length of stay still requires more in-depth studies. Similarly, the role of cultural values and positive or negative linguistic politeness strategies that TIC staff and international tourists might share in the airport contexts of interaction, as well as in other contexts, would be interesting areas of further studies.

## 7. Patents

There are no patents resulting from the work reported in this manuscript.

**Funding:** This research was funded by The Indonesian National Competitive Research Grant (Penelitian Dasar Kompetitif Nasional) grant number 117/ES/PG.02.00.PT/2022. The APC was funded by this grant.

**Informed Consent Statement:** Informed consent was obtained from all subjects involved in the study.

**Data Availability Statement:** No data availability.

**Acknowledgments:** The author would like to extend his thanks to the anonymous reviewers for their careful reading of the manuscript and for their insightful comments.

**Conflicts of Interest:** The author declares no conflict of interest. The funders had no role in the design of the study; in the collection, analyses, or interpretation of data; in the writing of the manuscript; or in the decision to publish the results.

## Appendix A. Interview Questions

1.　What are the main agenda of Lombok tourism?
2.　What are the strengths of Lombok tourism compared to others?
3.　What main issues are currently facing Lombok tourism?

4. What roles do the TIC and TIC staff play in the tourism agenda and issues?
5. How will these roles be improved in the future?

**Appendix B. Questionnaire**

Thick ($\sqrt{}$) to the most relevant answer to each question.

1. Are you one of the authorities in Lombok tourism?

☐ Yes ☐ No **Go to question #6**

2. What do you think of Lombok tourism in the last two years?

☐ Excellent
☐ Good
☐ As usual
☐ Quiet
☐ Very quiet

3. What do you think the government should do?

☐ More promotion of tourism objects
☐ More tourism events
☐ More cultural attractions
☐ More food and beverage festivals
☐ Others and please specify

4. How do you think Tourist Information Center (TIC) has performed their jobs?

☐ Excellent
☐ Very Good
☐ Good
☐ Bad
☐ Very Bad

5. In what aspect do you think TIC should improve? Please describe your answer.

6. Are you one of the TIC staff?

☐ Yes ☐ No **Go to question #11**

7. How do you think TIC staffs have done their jobs in the last two years?

☐ Excellent
☐ Good
☐ As usual
☐ Quiet
☐ Very quiet

8. What do you think the TIC staffs should develop?

☐ More pro-active in inviting tourist clients

☐ More informative to the tourist clients

☐ Learn more about the languages of the clients

☐ Obtain eye-catching spaces in the airport

☐ Others and please specify ………………………………

9. How have you performed your job at TIC?

☐ Excellent

☐ Very Good

☐ Good

☐ Bad

☐ Very Bad

10. In what aspect do you think you need to improve as a TIC agent? Please describe your answer.

```

```

11. Are you a visitor (tourist) to Lombok?

☐ Yes    ☐ No    **Go to question #16**

12. How helpful were the TIC staffs to you?

☐ Very helpful

☐ Helpful

☐ Average

☐ Helpless

☐ Very helpless

13. How useful was the information provided by the TIC staffs to you?

☐ Very useful

☐ Useful

☐ Average

☐ Useless

☐ Very useless

14. How do you think the information service provided by the TIC staff?

☐ Excellent

☐ Very Good

☐ Good

☐ Bad

☐ Very Bad

15.    What do you think of the following statement: The TIC service have supplied information I need for traveling around Lombok.

☐ Strongly agree

☐ Agree

☐ Not sure

☐ Disagree

☐ Strongly disagree

16.    Thank you very much for participating in the study.

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
