# Peer review of "English and Co-Construction of Solidarity between Language Agents and Tourists in Tourism Information Service"

_languages, doi:10.3390/languages8020126_

Round 1
Reviewer 1 Report
This article deals with an interesting topic (the role of English in tourism interactions and its link with solidarity) in a context which has not been widely studied, that of Lombok, Indonesia. The data collection and fieldwork that underpin the study are clearly very comprehensive. The conclusions from the study could be interesting and pertinent but, in my view, there are a number of points throughout the article that make either the argumentation or conclusions seem unclear. For these reasons, the article sometimes feels like that it is lacking in solidity. In my opinion, if the following issues can be addressed, the article could provide a valuable contribution to (socio)linguistic knowledge on situations of tourism:
- There seems to be a general assumption that any non-Indonesian tourist is likely to be English-speaking. This is very possibly true, but it might be helpful to provide some idea, based on the study data, as to whether this is actually the case.
- The author makes reference to Durkheim while discussing global community membership. This is no doubt interesting but the author sometimes makes it sound like Durkheim wrote on this exact subject himself rather than it being a case of the author (pertinently) mobilising Durkheim’s work in this new context. The same goes for Tönnies (and there is some variation in the spelling of this last name).
- Given the topic and the findings, I think the author might find it useful to explore some work on tourism situations (and especially tourism interactions) by Adam Jaworski and Crispin Thurlow, Alexandre Duchêne and Shuang Gao.
- Given its importance to the rationale behind the article, a little more discussion of what constitute “emergency tourism airport situations” would be welcome.
- I think it would be useful to explain in a little more detail what is meant by “introspective, retrospective and prospective interviews”. Similarly, the notions of “symbolic”, “organic” and “mechanical” solidarity could have been explained in more detail.
- What language(s) were the interviews conducted in? It would be useful to know if we are reading direct transcriptions or translations of these encounters.
- I’m not sure I follow the analysis of Extract 1: why does the author conclude that lines 13-15 are examples of language commonly used only among friends? What would be an equivalent exchange in English among strangers?
- With regards to speech accommodation and speech rate (pages 11-12), how did the author establish a “normal” speech rate? In other words, when participants are judged to accelerate or slow down their speech, what rate is this in relation to? The way I understand it, the base rate is set as an average of all the interactions recorded. But is it possible to infer from this base whether individual speakers actually speed up or slow down their speech? This information is possibly in the paragraphs that deal with this topic but, if so, it is not easily identifiable. In my opinion, clarification/rewording is probably necessary here.
- Similar comment regarding convergence towards speech styles. How does the author define when someone is converging in the recorded interaction? How do we know that this isn’t simply an idiolectal difference concerning the speaker(s) in question? On this topic, what evidence does the author have to suggest that it is unwanted behaviour from the tourists’ perspective?
- Some conclusions seem a little quick. For instance, the use of constructions such as “bro” and “sis” as signifying the construction of familial relationships and thus symbolic solidarity. This may indeed be the case, but I think a bit more (empirically based) discussion would be helpful to strengthen the argumentation. Otherwise, it seems more likely that it is linguistic accommodation due to the common linguistic practices mentioned in the paragraph following extract 5.
- Is there any empirical evidence for the practices studied here having “enhanced the quality of tourism handling” (p15)? Or is this the impression of the author? Either way, this conclusion needs to be better explained in my view.
---
In terms of the formal aspects, in my view, the structure of the article could be lightly revised. The introduction is very long and would benefit, in my opinion, from being broken up into different topic-based sections. For instance, what seems to be a literature review (pages 4-6) comes under this heading of “introduction” but would benefit from being a clearly identifiable section of its own. There is quite a bit of repetition between the introduction, analysis, discussion and conclusion; close proofreading could help streamline the article.
Not all of the references are formatted in the same way.
The English language expression is generally acceptable but there are quite a lot of non-standard formulations in English. These do not always cause problems for comprehension (although sometimes this is the case) but may need to be addressed, depending on the language policy of the journal. I have not undertaken close proofreading of the English, but it might be a good idea for the final version. There are a couple of English language points that do hinder understanding:
- p1 line 7-8: what does ‘identifying’ mean in this context?
-p5 lines 154-156: the reference to Green et al suggests that the findings are similar to those of the studies mentioned previously, but the sentence seems to suggest the opposite. Requires clarification/rewording.
Author Response
REPLY TO REVIEWER 1
Comments and Suggestions for Authors
This article deals with an interesting topic (the role of English in tourism interactions and its link with solidarity) in a context which has not been widely studied, that of Lombok, Indonesia. The data collection and fieldwork that underpin the study are clearly very comprehensive. The conclusions from the study could be interesting and pertinent but, in my view, there are a number of points throughout the article that make either the argumentation or conclusions seem unclear. For these reasons, the article sometimes feels like that it is lacking in solidity. In my opinion, if the following issues can be addressed, the article could provide a valuable contribution to (socio)linguistic knowledge on situations of tourism:
- There seems to be a general assumption that any non-Indonesian tourist is likely to be English-speaking. This is very possibly true, but it might be helpful to provide some idea, based on the study data, as to whether this is actually the case.
The use of “English-speaking tourist” or “English speaking tourists” has led to this mistake and these expressions in the revised text have been replaced in the revised text with a general term “international tourists”.
- The author makes reference to Durkheim while discussing global community membership. This is no doubt interesting but the author sometimes makes it sound like Durkheim wrote on this exact subject himself rather than it being a case of the author (pertinently) mobilising Durkheim’s work in this new context. The same goes for Tönnies (and there is some variation in the spelling of this last name).
That’s true. Durkheim and Töonies respectively talked about solidarity in workplace relations and in communities in general and not about tourism or interactions in tourism world. The way they have been quoted has been changed in the revised text giving more emphasis on the fact that their concepts and opinions have been borrowed, followed, or extended to agent-tourist airport interactions. Expressions like “following Durkheim (1994)” has been used in the revised version of the manuscript. Non-English variations of names have also been used in the revised version: “Toonies” has been replaced with “Töonies” and “Duchene” has been corrected with “DuchÄ›ne”.
- Given the topic and the findings, I think the author might find it useful to explore some work on tourism situations (and especially tourism interactions) by Adam Jaworski and Crispin Thurlow, Alexandre Duchêne and Shuang Gao.
The relevant works of the named authors have been integrated in the theoretical review and the discussion sections of the revised manuscript. The works of Jaworski and Thurlow on language and tourism habitus (Jaworksi & Thurlow, 2010) and tourism genre ((Thurlow & Jaworski, 2003) have also been incorporated in the revised manuscript. The works of DuchÄ›ne with colleagues (Heller & Pujolar) on language as tourism commodity has also been integrated (for example, Heller & DuchÄ›ne, 2012; Heller & DuchÄ›ne, 2016; Heller, Pujolar & DuchÄ›ne, 2014). Gao’s works on a tourism center in China and use of this center as a resource for learning English (see Gao, 2016; Gao, 2017;Gao, 2018) have also been included in the revised version of the manuscript.
- Given its importance to the rationale behind the article, a little more discussion of what constitutes “emergency tourism airport situations” would be welcome.
The “emergency” situations of an airport have been added to the revised manuscript (see page 4 paragraph 2).
- I think it would be useful to explain in a little more detail what is meant by “introspective, retrospective and prospective interviews”. Similarly, the notions of “symbolic”, “organic” and “mechanical” solidarity could have been explained in more detail.
Introspective, retrospective and prospective interviews have been explained in section 3 Materials and Methods (see page 7, paragraph 1) and this explanation has also been made with reference to the interview questions in Appendices A and B.
Explanation to organic and mechanical solidarity has also been added to the revised manuscript in a number of sections. Examples of organic solidarity can be seen in section 2 Literature Review page 5 (paragraph 1), section 4 Findings page 11 (paragraph 1), page 14 (paragraph 1), and elsewhere. Mechanical solidarity can be found in section 2 Literature Review page 5 (paragraph 1), section 4 Findings page 11 (paragraph 1), page 12 (paragraph 1), page 14 (paragraph 1), page 17 (paragraph 1) and elsewhere. I believe that these explanations and examples could provide the details required.
- What language(s) were the interviews conducted in? It would be useful to know if we are reading direct transcriptions or translations of these encounters.
The interview with the staff and the tourism officers were in Bahasa Indonesia and with the tourist respondents in English. This has been stated in the revised version section 3 Materials and Methods on page 7 paragraph 2.
- I’m not sure I follow the analysis of Extract 1: why does the author conclude that lines 13-15 are examples of language commonly used only among friends? What would be an equivalent exchange in English among strangers?
Explanation and equivalent examples have been added to the revised manuscript on page 12, paragraph 1.
- With regards to speech accommodation and speech rate (pages 11-12), how did the author establish a “normal” speech rate? In other words, when participants are judged to accelerate or slow down their speech, what rate is this in relation to? The way I understand it, the base rate is set as an average of all the interactions recorded. But is it possible to infer from this base whether individual speakers actually speed up or slow down their speech? This information is possibly in the paragraphs that deal with this topic but, if so, it is not easily identifiable. In my opinion, clarification/rewording is probably necessary here.
Normal speech rate in L1-L1 interactions is calculated based on the number of words produced in L1-L1 conversations divided by the length of the conversations in minutes. With 35 L1 speakers, 11727 words, and 69 minute interactions recorded, I found that the average speech rate is around 170 wpm. The speech rate in L1-L2 interactions is also calculated in the same way. With 4 L2 and 35 L1 speakers, 151151 words, and 1057 minute conversations, the average speech rate is 143 wpm.
This explanation has been added to the revised version of the manuscript on page 13 paragraph 5.
- Similar comment regarding convergence towards speech styles. How does the author define when someone is converging in the recorded interaction? How do we know that this isn’t simply an idiolectal difference concerning the speaker(s) in question? On this topic, what evidence does the author have to suggest that it is unwanted behaviour from the tourists’ perspective?
Phonological convergence can be seen in the speaker’s shifting phonologically from one phonological variety of English to another. Identifying the usual way a speaker(s) speak can help us determine shifts in the phonological variety of English being used. These shifts are patterned and predictable and linguistically known to relate to ethnolinguistic identity of the speaker. Clyne, Eisikovits, and Tollfree (2001) have reported that 55% Australian and New Zealand speakers of English are speakers of broad Australian accents where certain types of vowels are added glides resulting in diphthongs. This information has been added to the revised manuscript and examples of these ethnolects are also shown in the revision.
- Some conclusions seem a little quick. For instance, the use of constructions such as “bro” and “sis” as signifying the construction of familial relationships and thus symbolic solidarity. This may indeed be the case, but I think a bit more (empirically based) discussion would be helpful to strengthen the argumentation. Otherwise, it seems more likely that it is linguistic accommodation due to the common linguistic practices mentioned in the paragraph following extract 5.
Empirical data for more discussion on the issue and elaborating more on the use of the kinship terms might strengthen the argument, but this, in its nature, a new topic of investigation. Nonetheless, the case is existent only in interactions of tourists with roughly the same age as the staff. The use of such terms is common among that generation and associating them with linguistic accommodation is more plausible. .
- Is there any empirical evidence for the practices studied here having “enhanced the quality of tourism handling” (p15)? Or is this the impression of the author? Either way, this conclusion needs to be better explained in my view.
Yes, there is no empirical evidence for it so a more tentative claim (using ‘might have increased’) has been made in the revised manuscript. It has also suggested the case to be worthy of further research. Thus, the conclusion has been changed following the suggestion.
---
In terms of the formal aspects, in my view, the structure of the article could be lightly revised. The introduction is very long and would benefit, in my opinion, from being broken up into different topic-based sections. For instance, what seems to be a literature review (pages 4-6) comes under this heading of “introduction” but would benefit from being a clearly identifiable section of its own. There is quite a bit of repetition between the introduction, analysis, discussion and conclusion; close proofreading could help streamline the article.
A new section (section 2 Literature Review) has been added, following the suggestion..
Not all of the references are formatted in the same way.
Now, all the references have been formatted following the journal’s template.
The English language expression is generally acceptable but there are quite a lot of non-standard formulations in English. These do not always cause problems for comprehension (although sometimes this is the case) but may need to be addressed, depending on the language policy of the journal. I have not undertaken close proofreading of the English, but it might be a good idea for the final version. There are a couple of English language points that do hinder understanding:
All mistakes in the draft have been corrected in the revised version of the manuscript. Proofreading has also been assisted by an L1 speaker of English.
- p1 line 7-8: what does ‘identifying’ mean in this context?
“[I]dentifying” has been replaced with “explicating” as it is more appropriate with the purpose of the article.
-p5 lines 154-156: the reference to Green et al suggests that the findings are similar to those of the studies mentioned previously, but the sentence seems to suggest the opposite. Requires clarification/rewording.
Thank you. I have used a wrong adjective “detrimental” and what I meant to say was “determinant”. Correction has been made.
I also appreciate this critical and comprehensive review and the article is in better shape after the comments and the recommendations have been integrated. Thank very much indeed.

Reviewer 2 Report
The article is very clear about it aims and objectives. The methodological approach is evident and well supported with examples. I would recommend to the author only to specify better the choice of the interviews extracts s/he made and why they are central in understanding his/her analysis. The structure of the analysis is clear and well written. I would recommend a native speaker reading because there are minor non-native ways of writing in English (the tourism language services for example instead of tourism language services). Also I would recommend to always choose a more formal way of writing (this study..... it is possible to... and not the use of first person pronoun as at the end of section 3 'I can compare').
The bibliography is updated
Author Response
REPLY TO REVIEWER 2
Comments and Suggestions for Authors
The article is very clear about it aims and objectives. The methodological approach is evident and well supported with examples. I would recommend to the author only to specify better the choice of the interviews AND extracts s/he made and why they are central in understanding his/her analysis.
The interviews and the extracts have been carefully selected from samples of similar cases in relation to the arguments put forward in the analysis. The relevance to the argument is the only reason for the selection. The recommendation has actually been implemented in the manuscript.
The structure of the analysis is clear and well written. I would recommend a native speaker reading because there are minor non-native ways of writing in English (the tourism language services for example instead of tourism language services).
The minor non-native English expressions were due to careless editing in the previous version of the article so that many grammatical mistakes and misspellings can be found in the manuscript. These mistakes have been corrected, following the suggestions made by other reviewers and by the author’s more careful reading and re-editing of the revised manuscript.
Also I would recommend to always choose a more formal way of writing (this study..... it is possible to... and not the use of first person pronoun as at the end of section 3 'I can compare').
Formal way of writing has been followed and the use of 1st pronoun has been replaced with impersonal subjects (for example, “the study finds that …” or “let us compare…”) or passive voices (example, “the reasons were identified by …”).
The bibliography is updated
Thank you very much. New references have been added in the revised manuscript following new ideas and reference works recommended by other reviewers.
I do appreciate these thoughtful comments and I have followed them thoroughly for the betterment of the manuscript quality. Thank you very much indeed.

Reviewer 3 Report
The article entitled "English and Co-Construction of Solidarity between Language Agents and Tourists in Tourism Information Services", seems to be a very appealing and innovative research within the context of the tourism industry from a linguistic perspective. It goes without saying that many research works abound in the literature regarding the importance of the persuasive rhetorical function of communication strategies in different written and spoken genres in English as well as in English in comparison with other languages within the field of tourism. Nonthistanding this, the present research focuses on a new topic that, as far as I know, does not exist in the current literature: the use of the English language as a lingua franca and how social solidarity in international airports is co-constructed between language agents and tourists in Tourism Information Services.
Regarding the formal aspects of the text, I guess the text is very well written in English and, in general, follows the academic conventions of a research paper: Introduction, Corpus description and method of analysis, Findings, Discussion, Conclusions and References. Likewise, it is worth pointing out how the author/s uses/use transition paragraphs at the end of each paragraph to guide readers from the very beginning of the work. Despite this, it would be recommendable to use more softeners markers in the discussion section, as sometimes the interpretaion given to the findings obtained appear to be a bit imposing. In a similar vein, as the objetive pursued in this research work is clearly explicit in the introduction section, there is no need, in my view, to repeat it several times throughout the text. The Chi-square analysis used to obtain statistical data in the speech accommodation variable must also be mentioned in the methodology section.
Regarding the content of this research, I miss in the introduction section some research questions. Although the objective and the gap found by the author/s are clearly stated, it would be convenient to formulate two or three research questions in the introductory section. As regards symbolic solidarity, the author/s points/point out that "Honesty, reciprocity and mutuality in the service encounter can lead to symbolic solidarity between the interactants and this can be described as mechanical and organical solidarity". In spite of the fact that the both types of solidarity are vey well defined and explained, I feel it would be advisable to use in brackets "rapport vs report" to faciliate a better understanding of both terms. As for the linguistic features of solidarity, the research only draws its attention towards style accommodation, code switching and the use of kin terms. As lingustic solidarity can considered very similar to "positive politeness strategies", I wonder why the author/s has/have not included other strategies related to positive politeness (See Brown and Levinson). In this regard, it would be highly recommendable to make reference to linguistic politeness in the theoretical background. In addition, I hold the opinion that the author/s should include some of the results drawn from the questionnaire distributed among tourists so as to really know how tourists feel after being assisted by agents. I wonder whether excessive use of solidarity markers could make tourists feel uncomfortable. We must take into consideration that tourists come from different cultural background and we cannot overlook the fact that cultural values impact on the way individuals communicate and interact with each other. That is the reason why I think a more thorough analysis must be carried out in this research by considering politeness theory (Brown and Levinson, Scollon and Scollon, etc.) , cultural dimensions and values (e.g. Geert Hofstede) and the results drawn from the questionnaire. In this way, the findings could be more feasible.
If the author/s took into account the aforementioned recommendations and advices, I think the research could be accepted for publication.
Reviewer 4 Report
Typos are highlighted in the file

Round 2
Reviewer 1 Report
The author has worked hard to revise the manuscript but, in my view, the changes remain relatively superficial and the main concerns expressed with regards to my previous "major revisions" evaluation have not been dealt with. The author has provided some elements of contextualisation in the author's response, but these elements are not enough to convince me that the issues do no need to addressed in the article itself. In my view, it is absolutely necessary for the following elements to be adjusted in the article before it can be deemed publishable:
- Some analyses seem to be based on a theoretical "what might have happened" situation, rather than actual data. For example, the discussions on formal/informal greetings on page 11. Is there any data to suggest that the participants found the particular greetings used informal?
- I still don't see how the quantitative analyses of speech accommodation work. As far as I can tell, even though this is now very clearly explained, it is still entirely possible for the differences to be down to individual ideolectal differences, rather than speech accommodation. I would advise focusing the argument on the qualitative data that follows because these data do provide empirical sources to support the author's argument.
- The same thing applies to discussion of English language variation. What data are there to support that a certain participant felt "displeasure"? Also, this section seems potentially contradictory: the author argues that symbolic solidarity was produced because the intention was good, but it seems that displeasure was still felt. Can we really talk of solidarity in such a situation? I feel that a more nuanced discussion would be beneficial.
Aside from these points, the new changes lead me to the following observations:
- It is not clear exactly what is meant by "ideological reasons" in section 4.1
- I would advise breaking page 4 up into different paragraphs because there is a huge variety of themes discussed in this piece of text and it is hard to follow
- I think it might be necessary to sometimes rethink how the new elements of the theoretical background are used. For example, I don't recall Jaworski and Thurlow saying that travel is psychologically depressing. I may be wrong but, if so, a direct quote would be better. Similarly, I'm not sure where Blommaert (2005) talks about the following: "travellers are in constant nervousness about how they might construct themselves in the visited country, how the host communities might treat them, what they can or cannot do there, and whether their sociolinguistic repertoires could help them in the new situations". Again, I may be wrong but I have no recollection of Blommaert discussing tourism in that particular book. If I am wrong, a direct quote would be helpful.
- On a related note, it is not always clear how these new theoretical elements link to the present study. For example, there is a lot of discussion of objects being commodified but it is not immediately clear which objects in this particular context are concerned and this commodification process does not appear in the data. Is this concept useful to the present study? If so, this needs to be made explicit.
Finally, the English is very often non-standard (especially in terms of grammatical structure). This very rarely impedes understanding and, personally, I have no issue with it, but I prefer to mention it here as I do not know what the language policy of the journal is with regards to this question.
For all of the above reasons, I still believe this article needs major revision before it can be published.
Author Response
ROUND 2: REPLY TO REVIEWER 1
Review Report Form
Open Review
(x) I would not like to sign my review report
( ) I would like to sign my review report
Quality of English Language
( ) English very difficult to understand/incomprehensible
( ) Extensive editing of English language and style required
(x) Moderate English changes required
( ) English language and style are fine/minor spell check required
( ) I am not qualified to assess the quality of English in this paper
Yes |
Can be improved |
Must be improved |
Not applicable |
|
Is the content succinctly described and contextualized with respect to previous and present theoretical background and empirical research (if applicable) on the topic? |
( ) |
(x) |
( ) |
( ) |
Are all the cited references relevant to the research? |
( ) |
(x) |
( ) |
( ) |
Are the research design, questions, hypotheses and methods clearly stated? |
(x) |
( ) |
( ) |
( ) |
Are the arguments and discussion of findings coherent, balanced and compelling? |
( ) |
(x) |
( ) |
( ) |
For empirical research, are the results clearly presented? |
( ) |
( ) |
(x) |
( ) |
Is the article adequately referenced? |
(x) |
( ) |
( ) |
( ) |
Are the conclusions thoroughly supported by the results presented in the article or referenced in secondary literature? |
( ) |
(x) |
( ) |
( ) |
Comments and Suggestions for Authors
The author has worked hard to revise the manuscript but, in my view, the changes remain relatively superficial and the main concerns expressed with regards to my previous "major revisions" evaluation have not been dealt with. The author has provided some elements of contextualisation in the author's response, but these elements are not enough to convince me that the issues do no need to addressed in the article itself. In my view, it is absolutely necessary for the following elements to be adjusted in the article before it can be deemed publishable:
- Some analyses seem to be based on a theoretical "what might have happened" situation, rather than actual data. For example, the discussions on formal/informal greetings on page 11. Is there any data to suggest that the participants found the particular greetings used informal?
REPLY: References to tape script and line numbers have been added to the manuscript in order to show that the data were in fact from empirical data and not from theoretical ones (see page 11 paragraph 1). Informality of the greetings has been illustrated and samples of formal greetings have also been presented on page 12 (paragraph 1),
- I still don't see how the quantitative analyses of speech accommodation work. As far as I can tell, even though this is now very clearly explained, it is still entirely possible for the differences to be down to individual ideolectal differences, rather than speech accommodation. I would advise focusing the argument on the qualitative data that follows because these data do provide empirical sources to support the author's argument.
REPLY: The quantitative analyses have been presented in response to a reviewer’s comment in the previous round asking for more convincing generalizable conclusion on speech accommodation as evidence of solidarity. The issue of the speech accommodation being individual ideolects has been addressed in the revised manuscript highlighting the fact that the data were drawn from more than 80 hours of conversations involving 35 speakers in various airport contexts. Individual ideolects might have been controlled or moderated in the data. This information has been added to the revised manuscript on page 14 paragraph 2. These quantitative findings have also been emphasized with qualitative findings where data from retrospective interviews with the participants have been presented to put more evidences for the case of speech accommodation (see page 14 paragraph 2).
- The same thing applies to discussion of English language variation. What data are there to support that a certain participant felt "displeasure"? Also, this section seems potentially contradictory: the author argues that symbolic solidarity was produced because the intention was good, but it seems that displeasure was still felt. Can we really talk of solidarity in such a situation? I feel that a more nuanced discussion would be beneficial.
REPLY: “Displeasure” is not the right word for the tourist’s feeling in the context. It is too strong for it. Instead, the tourist was worried about or concerned with how the staff might have taken what he said given the accent that he had used and this concern prompted him to switch to international variety of English which is generally understood by non-native speakers like Danti the TIC staff. This issue was brought up in the recording by the tourist when he asked Danti the staff how she acquired the Australian accent. The latter reported to have learned it from frequent contact in the job with Australians and used it to associate herself with Australian English and Australian tourists, whom she said are ‘ramah’ [friendly] (see page 15 paragraph 2).
Aside from these points, the new changes lead me to the following observations:
- It is not clear exactly what is meant by "ideological reasons" in section 4.1
REPLY: Definition of “ideology’ has been explicitly made in the revised manuscript as patterned beliefs or ideas about how tourism should be better served as one of important public services. The term “Ideological reasons” is meant to refer to the political, administrative, economic and cultural rationales why TIC services should be established at LIA the airport. This information has been added to the revised manuscript on page 8 paragraph 3.
- I would advise breaking page 4 up into different paragraphs because there is a huge variety of themes discussed in this piece of text and it is hard to follow.
REPLY: yes, suggestion has been followed. Actually, merging them was a mistake unidentified on the computer screen due to dominant appearance of track changes. Information has been added and the paragraph has been broken into 3 paragraphs (see page 3-4).
- I think it might be necessary to sometimes rethink how the new elements of the theoretical background are used. For example, I don't recall Jaworski and Thurlow saying that travel is psychologically depressing. I may be wrong but, if so, a direct quote would be better. Similarly, I'm not sure where Blommaert (2005) talks about the following: "travellers are in constant nervousness about how they might construct themselves in the visited country, how the host communities might treat them, what they can or cannot do there, and whether their sociolinguistic repertoires could help them in the new situations". Again, I may be wrong but I have no recollection of Blommaert discussing tourism in that particular book. If I am wrong, a direct quote would be helpful.
REPLY: You are right. I have misunderstood what Jaworski and Thurlow (2010) write. They perceive tourism as positively motivated by pleasure with tourism experiences and commodities and the displacement of tourists and their languages will affect to visited people and their languages. in such contexts, various formulaic informal forms of tourist greetings or farewells can be seen as linguistic forms of mechanical solidarity in tourist-host interactions.
Blommaert (2005: 203) did not write about tourism but mobility and, I believe, touring is one example of human mobility. Blommaert predicted 3 things that travelers need to ask themselves when moving to new places: “(a) what people can or cannot do … [in the visited spaces], (b) the value and function of their sociolinguistic repertoires; (c) their identities, both self-constructed (inhabited) and ascribed by others”. Extending on these ideas, a word too strong for the expression has been used suggesting the traveling tourists are “nervous” with the situation and need help to overcome their “nervousness” and, certainly, it is not the case. In the revised manuscript, the strength of the expression has been reduced to the act of self-questioning or considering what might happen rather than being nervous.
- On a related note, it is not always clear how these new theoretical elements link to the present study. For example, there is a lot of discussion of objects being commodified but it is not immediately clear which objects in this particular context are concerned and this commodification process does not appear in the data. Is this concept useful to the present study? If so, this needs to be made explicit.
REPLY: The main focus of the article is on solidarity. Commodity or commodification of English or others is not the issue at least in the current stage of the study. It is going to be for another article.
Finally, the English is very often non-standard (especially in terms of grammatical structure). This very rarely impedes understanding and, personally, I have no issue with it, but I prefer to mention it here as I do not know what the language policy of the journal is with regards to this question.
REPLY: The grammatical structure has been revisited and a native speaker of English is now reading the current draft for clarity of ideas and grammar check.
For all of the above reasons, I still believe this article needs major revision before it can be published.
REPLY: Thank you very much. The reviews and the comments are very much appreciated and they have shaped the article in a better form. Thank you very much indeed.
Submission Date
31 December 2022
Date of this review
12 Mar 2023 17:01:02
Reviewer 3 Report
Having read the revised version of the manuscript, I have just checked out that the author/s has/have included all the changes I recommended in my first report. Despite this, in the conclusion section, I would recommend to include the correlation between cultural values (from an anthropological perspective) and politeness linguistic strategies (positive vs negative) as a possible area of research for the near future. On the whole, I think that the article could be published if the aforementioned minor revision was carried out.
Author Response
Review Report Form
Open Review
( ) I would not like to sign my review report
(x) I would like to sign my review report
Quality of English Language
( ) English very difficult to understand/incomprehensible
( ) Extensive editing of English language and style required
( ) Moderate English changes required
(x) English language and style are fine/minor spell check required
( ) I am not qualified to assess the quality of English in this paper
Yes |
Can be improved |
Must be improved |
Not applicable |
|
Is the content succinctly described and contextualized with respect to previous and present theoretical background and empirical research (if applicable) on the topic? |
(x) |
( ) |
( ) |
( ) |
Are all the cited references relevant to the research? |
(x) |
( ) |
( ) |
( ) |
Are the research design, questions, hypotheses and methods clearly stated? |
(x) |
( ) |
( ) |
( ) |
Are the arguments and discussion of findings coherent, balanced and compelling? |
(x) |
( ) |
( ) |
( ) |
For empirical research, are the results clearly presented? |
(x) |
( ) |
( ) |
( ) |
Is the article adequately referenced? |
(x) |
( ) |
( ) |
( ) |
Are the conclusions thoroughly supported by the results presented in the article or referenced in secondary literature? |
( ) |
(x) |
( ) |
( ) |
Comments and Suggestions for Authors
Having read the revised version of the manuscript, I have just checked out that the author/s has/have included all the changes I recommended in my first report. Despite this, in the conclusion section, I would recommend to include the correlation between cultural values (from an anthropological perspective) and politeness linguistic strategies (positive vs negative) as a possible area of research for the near future. On the whole, I think that the article could be published if the aforementioned minor revision was carried out.
REPLY: The recommendation has been followed and the need to study the role of cultural values and positive and negative politeness strategies in tourist-host airport contexts and others has been added in the conclusion section of the article (see page 18 paragraph 4, the last sentence).
Submission Date
31 December 2022
Date of this review
19 Mar 2023 19:15:40
Round 3
Reviewer 1 Report
The author has made some effort to address some of my previous concerns, both in the manuscript and by justifying them through the author reply. I am of the opinion that these changes remain relatively superficial. Thus, almost all of my concerns from the previous report remain.
With a few modifications based upon my previous suggestions, this article could, in my opinion, be accepted for publication. However, given that the revised manuscript is very similar to the previous version, I prefer to let the journal editorial committee take a decision (and I have contacted them to explain this).
I don't believe the Blommaert (2005) reference is correct, I think the work being referenced is Blommaert, Collins & Slembrouck (2005), "Spaces of multilingualism", Language and Communication 25, 197-216. Also, another reference to Blommaert (2017) appears in the reference list but is not included in the text.
Author Response
ROUND 3: REPLY TO REVIEWER 1
In round 3, Reviewer 1 wrote
I don't believe the Blommaert (2005) reference is correct, I think the work being referenced is Blommaert, Collins & Slembrouck (2005), "Spaces of multilingualism", Language and Communication 25, 197-216. Also, another reference to Blommaert (2017) appears in the reference list but is not included in the text.
New Revision: In the new revised version, reference to Blommaert (2005) and Blommaert (2017) has been revisited. Blommaert’s (2005) notion of uneven distribution, Blommaert, Collins and Slembrouck’s (2005) scalar nature of linguistic repertoires, and Blommaert’s (2017) argument for social, ideological and moral nature of contexts have been added the ideological reasons why the airport situation under study is emergent in nature and it, consequently, requires the language service provided by the TIC staff. These references have been included both in the text and in the reference.
Reviewer 1 also wrote that “all most all of my concerns from the previous report remain”. This remark is taken to mean that all of the concerns should be looked at once again and accordingly responded to. This is presented below.
Review Report Form
Open Review
(x) I would not like to sign my review report
( ) I would like to sign my review report
Quality of English Language
( ) English very difficult to understand/incomprehensible
( ) Extensive editing of English language and style required
(x) Moderate English changes required
( ) English language and style are fine/minor spell check required
( ) I am not qualified to assess the quality of English in this paper
Yes |
Can be improved |
Must be improved |
Not applicable |
|
Is the content succinctly described and contextualized with respect to previous and present theoretical background and empirical research (if applicable) on the topic? |
( ) |
(x) |
( ) |
( ) |
Are all the cited references relevant to the research? |
( ) |
(x) |
( ) |
( ) |
Are the research design, questions, hypotheses and methods clearly stated? |
(x) |
( ) |
( ) |
( ) |
Are the arguments and discussion of findings coherent, balanced and compelling? |
( ) |
(x) |
( ) |
( ) |
For empirical research, are the results clearly presented? |
( ) |
( ) |
(x) |
( ) |
Is the article adequately referenced? |
(x) |
( ) |
( ) |
( ) |
Are the conclusions thoroughly supported by the results presented in the article or referenced in secondary literature? |
( ) |
(x) |
( ) |
( ) |
Comments and Suggestions for Authors
The author has worked hard to revise the manuscript but, in my view, the changes remain relatively superficial and the main concerns expressed with regards to my previous "major revisions" evaluation have not been dealt with. The author has provided some elements of contextualisation in the author's response, but these elements are not enough to convince me that the issues do no need to addressed in the article itself. In my view, it is absolutely necessary for the following elements to be adjusted in the article before it can be deemed publishable:
- Some analyses seem to be based on a theoretical "what might have happened" situation, rather than actual data. For example, the discussions on formal/informal greetings on page 11. Is there any data to suggest that the participants found the particular greetings used informal?
REPLY: References to tape script and line numbers have been added to the manuscript in order to show that the data were in fact from empirical data and not from theoretical ones (see page 11 paragraph 1). Informality of the greetings has been illustrated and samples of formal greetings have also been presented on page 12 (paragraph 1),
New Revision: Data suggesting informal relationship between the host and the tourist in the extract have been explicated in the new revised manuscript where ‘bye” (in line 14 and 15) has been used as samples of informal greetings.
- I still don't see how the quantitative analyses of speech accommodation work. As far as I can tell, even though this is now very clearly explained, it is still entirely possible for the differences to be down to individual ideolectal differences, rather than speech accommodation. I would advise focusing the argument on the qualitative data that follows because these data do provide empirical sources to support the author's argument.
REPLY: The quantitative analyses have been presented in response to a reviewer’s comment in the previous round asking for more convincing generalizable conclusion on speech accommodation as evidence of solidarity. The issue of the speech accommodation being individual ideolects has been addressed in the revised manuscript highlighting the fact that the data were drawn from more than 80 hours of conversations involving 35 speakers in various airport contexts. Individual ideolects might have been controlled or moderated in the data. This information has been added to the revised manuscript on page 14 paragraph 2. These quantitative findings have also been emphasized with qualitative findings where data from retrospective interviews with the participants have been presented to put more evidences for the case of speech accommodation (see page 14 paragraph 2).
New Revision: The study involved 35 speakers in 80 hour interactions and 23808 words have been used as data for analysis. The analysis involves comparative inferential statistical analyses with significant accommodation of speech rate when tourists with L1 and L2 English speaker background speak with TIC agent with Lx background. The assumption that the data might be individual ways of speaking (or ideolects) has actually been refuted with the nonparametric inferential analysis as well as the representative number of samples (N > 35 although N=30 is usually the minimum number of samples required in parametric analysis). Besides, the figures used in the Chi-square analyses are also mean scores (or numbers) of words per minute produced by participants with L1, L2 and Lx English background in various settings: L1-L1, L1-L2, L1-Lx, L2-L2, L2-Lx and Lx-Lx. With these contextual varieties of data sources, the speech accommodation could have not been individual ways of speaking by the samples. This information has also been added to the new revised version of the manuscript.
- The same thing applies to discussion of English language variation. What data are there to support that a certain participant felt "displeasure"? Also, this section seems potentially contradictory: the author argues that symbolic solidarity was produced because the intention was good, but it seems that displeasure was still felt. Can we really talk of solidarity in such a situation? I feel that a more nuanced discussion would be beneficial.
REPLY: “Displeasure” is not the right word for the tourist’s feeling in the context. It is too strong for it. Instead, the tourist was worried about or concerned with how the staff might have taken what he said given the accent that he had used and this concern prompted him to switch to international variety of English which is generally understood by non-native speakers like Danti the TIC staff. This issue was brought up in the recording by the tourist when he asked Danti the staff how she acquired the Australian accent. The latter reported to have learned it from frequent contact in the job with Australians and used it to associate herself with Australian English and Australian tourists, whom she said are ‘ramah’ [friendly] (see page 15 paragraph 2).
New Revision: No change is made in response to this comment. The notion of displeasure has been replaced with the participant’s “concern” or “worry” with possible misunderstanding or miscommunication. We do not know the reviewer’s opinion about this revision.
Aside from these points, the new changes lead me to the following observations:
- It is not clear exactly what is meant by "ideological reasons" in section 4.1
REPLY: Definition of “ideology’ has been explicitly made in the revised manuscript as patterned beliefs or ideas about how tourism should be better served as one of important public services. The term “Ideological reasons” is meant to refer to the political, administrative, economic and cultural rationales why TIC services should be established at LIA the airport. This information has been added to the revised manuscript on page 8 paragraph 3.
New Revision: Ideology has been defined as above and we are not sure if the definition is accepted or not. Ideological reasons have also been defined using Blommaert’s (2005) cognitive definition of ideology. The definition might be superficial but given the limit of space for the article and the focus of the article, which is not on ideology, we cannot have defined them in a much deeper discussion to the degree that they should have been.
- I would advise breaking page 4 up into different paragraphs because there is a huge variety of themes discussed in this piece of text and it is hard to follow.
REPLY: yes, suggestion has been followed. Actually, merging them was a mistake unidentified on the computer screen due to dominant appearance of track changes. Information has been added and the paragraph has been broken into 3 paragraphs (see page 3-4).
New Revision: The suggestion has been followed and we believe that there will not be any problems with this.
- I think it might be necessary to sometimes rethink how the new elements of the theoretical background are used. For example, I don't recall Jaworski and Thurlow saying that travel is psychologically depressing. I may be wrong but, if so, a direct quote would be better. Similarly, I'm not sure where Blommaert (2005) talks about the following: "travellers are in constant nervousness about how they might construct themselves in the visited country, how the host communities might treat them, what they can or cannot do there, and whether their sociolinguistic repertoires could help them in the new situations". Again, I may be wrong but I have no recollection of Blommaert discussing tourism in that particular book. If I am wrong, a direct quote would be helpful.
REPLY: You are right. I have misunderstood what Jaworski and Thurlow (2010) write. They perceive tourism as positively motivated by pleasure with tourism experiences and commodities and the displacement of tourists and their languages will affect to visited people and their languages. in such contexts, various formulaic informal forms of tourist greetings or farewells can be seen as linguistic forms of mechanical solidarity in tourist-host interactions.
Blommaert (2005: 203) did not write about tourism but mobility and, I believe, touring is one example of human mobility. Blommaert predicted 3 things that travelers need to ask themselves when moving to new places: “(a) what people can or cannot do … [in the visited spaces], (b) the value and function of their sociolinguistic repertoires; (c) their identities, both self-constructed (inhabited) and ascribed by others”. Extending on these ideas, a word too strong for the expression has been used suggesting the traveling tourists are “nervous” with the situation and need help to overcome their “nervousness” and, certainly, it is not the case. In the revised manuscript, the strength of the expression has been reduced to the act of self-questioning or considering what might happen rather than being nervous.
New Revision:
With regards to this, Reviewer 1 wrote
I don't believe the Blommaert (2005) reference is correct, I think the work being referenced is Blommaert, Collins & Slembrouck (2005), "Spaces of multilingualism", Language and Communication 25, 197-216. Also, another reference to Blommaert (2017) appears in the reference list but is not included in the text.
New Revision: In the new revised version, the relevance of Blommaert’s (2005), Blommaert’s (2017) and Blommaert, Collins and Slembrouck’s (2005) ideas to the manuscript have been highlighted more. Blommaert’s (2005) notion of uneven distribution, Blommaert, Collins and Slembrouck’s (2005) scalar nature of linguistic repertoires, and Blommaert’s (2017) argument for social, ideological and moral nature of contexts have been integrated to the new revised manuscript.
- On a related note, it is not always clear how these new theoretical elements link to the present study. For example, there is a lot of discussion of objects being commodified but it is not immediately clear which objects in this particular context are concerned and this commodification process does not appear in the data. Is this concept useful to the present study? If so, this needs to be made explicit.
REPLY: The main focus of the article is on solidarity. Commodity or commodification of English or others is not the issue at least in the current stage of the study. It is going to be for another article.
New Revision: search for words like ‘commodity’ or ‘commoditized’ or ‘commodified’ in the new revised manuscript has been done and no words were found. We believe that looking at the airport interactions in the contexts under study here from linguistic commodification point of view would be an interesting research article and, perhaps, this is the next research agenda.
Finally, the English is very often non-standard (especially in terms of grammatical structure). This very rarely impedes understanding and, personally, I have no issue with it, but I prefer to mention it here as I do not know what the language policy of the journal is with regards to this question.
REPLY: The grammatical structure has been revisited and a native speaker of English is now reading the current draft for clarity of ideas and grammar check.
New Revision: a native speaker has read the manuscript and revision has also been made following his/her remarks.
For all of the above reasons, I still believe this article needs major revision before it can be published.
REPLY: Thank you very much. The reviews and the comments are very much appreciated and they have shaped the article in a better form. Thank you very much indeed.
New Revision: for the second time, comments of the 2nd round of review were revisited and more revisions have been made to the manuscript after rereading the comments from the 2nd round of review.
Submission Date
31 December 2022
Date of this review
03 Apr 2023 16:15:52
